# Structure of a stripped-down and tuned-up far-red phycobilisome
Giovanni Consoli [1,3], Ho Fong Leong [1,3], Geoffry A. Davis [1,2,3], Tom Richardson[1], Aiysha McInnes[1], James W. Murray [1], Andrea Fantuzzi [1] ✉ & A. William Rutherford [1] ✉

A diverse subset of cyanobacteria can transiently modify their photosynthetic machinery during far-red light photoacclimation to drive photosynthesis with less energetic photons (700 nm–800 nm). To achieve this, all the main light-driven components of the photosynthetic apparatus, including their allophycocyanin antenna, are replaced with red-shifted paralogues. Recent studies based on the structure of an incomplete complex provided some insights into the tuning of the far-red phycobiliproteins. Here, we solved the structure of the intact bicylindrical allophycocyanin complex from the cyanobacterium *Chroococcidiopsis thermalis* PCC 7203 at a resolution of 2.51 Å determined by Cryo-electron microscopy single particle analysis. A comparison between conserved structural features in far-red and white light allophycocyanin cores provides insight on the evolutionary adaptations needed to optimize excitation energy transfer in the far-red light adapted photosynthetic apparatus. The reduction in antenna size in far-red photosynthesis suggests a need to optimize membrane packing to increase the number of photosystems and tune the ratio between chlorophyll *f* molecules and bilin pigments, while the wider spread in the absorption range of the bilins suggests faster and more efficient excitation energy transfer to far-red Photosystem II by limiting backflow of excitation from the reaction centres to the far-red bilin pigments.

In some ecological niches, such as those found in cyanobacterial mats[1], porous rocks[2], limestone caves[3] and other shaded environments[4], the ratio between visible light (400–700 nm) and far-red light (700–800 nm) decreases drastically. In these conditions, a small but phylogenetically diverse group of cyanobacteria is capable of far-red light photoacclimation (FaRLiP)[5], extending the "red limit" of photosynthesis using photons up to 800 nm in wavelength[6].

During this process, a gene cluster of ~21 genes[7], coding for specialized far-red-light-adapted paralogues of all the main components of the photosynthetic apparatus, replaces their white light counterparts. The far-red paralogous subunits of Photosystem II (PSII) and Photosystem I (PSI) incorporate a small number of longer wavelength chlorophylls, Chlorophyll *d* (Chl *d*), and Chlorophyll *f* (Chl *f*) (~8–10% of total chlorophylls)[6,8] in specific positions, further redshifting their absorption spectra. On the other hand, the far-red paralogues of the phycobilisomes, the major light-harvesting antenna proteins in cyanobacteria, retain the same type of pigment, phycocyanobilin, but shift their absorption by modifying their protein environment[9–11].

Phycobilisomes are a structurally and functionally diverse family of light-harvesting complexes, which bind open-chain tetrapyrroles, known as bilins, that absorb visible light and funnel excitation energy to the photosystems. The extreme variability in size, arrangement and absorbance profile of phycobilisomes reflects the evolutionary adaptation of these antenna complexes to a number of ecological niches[12].

Given the small number of long-wavelength chlorophylls present in far-red-adapted photosystems[6], a red-shifted antenna, which is capable of efficiently funneling the excitation energy of far-red photons into the far-red photosystems, is necessary to increase the antenna size for wavelengths above 700 nm. Most FaRLiP gene clusters contain five phycobiliprotein paralogues (ApcB2, ApcD5, ApcD3, ApcD2, and ApcE2), that assemble with other subunits in common with white light phycobilisomes (ApcF and ApcC) into a bicylindrical complex[11], which is significantly smaller than all the other classes of phycobilisomes[12]. However, some species (e.g., *Chroococcidiopsis thermalis* PCC 7203) possess a paralogue of ApcF, named ApcF2, that has been suggested to replace ApcF during FaRLiP[13,14]. For some species, including

[1]Department of Life Sciences, Imperial College, London, UK. [2]Present address: Department of Plant Biochemistry, Biology, Ludwig Maximilian University of Munich, Planegg-Martinsried, Germany. [3]These authors contributed equally: Giovanni Consoli, Ho Fong Leong, Geoffry A. Davis. ✉e-mail: a.fantuzzi@imperial.ac.uk; a.rutherford@imperial.ac.uk

*C. thermalis* PCC 7203, the gene coding for ApcF2 is located outside the FaRLiP cluster.

Similar to white-light phycobilisome (WL-PBS) cores, in far-red allophycocyanin (FR-APC), α- (ApcD2, ApcD3, ApcD5) and β- (ApcB2) heterodimers (αβ) assemble into toroidal hexameric (αβ)₃ complexes[10,15,16] that are stacked with the help of ApcC onto a specialized core-membrane linker protein, ApcE2, that also anchors the complex to the stromal side of PSII[17]. Compared to the far-red light-adapted subunits, the chromophores in the β-subunits (ApcB2) appear to be structurally more similar to their WL version[10] and therefore have similar absorption properties. On the other hand, in the α-subunits the pyrrole ring A of each bilin pigment is more coplanar with its other rings, leading to a red-shifted absorption compared to WL α-subunits, a feature that has also been observed in heterologously expressed biliproteins[11,18].

In the bicylindrical FR-APC, the bilins contained in ApcE2 and ApcD3 have been suggested to act as terminal emitters[10]. In contrast to all other phycobiliproteins, these two subunits lack the cysteine that normally covalently binds the bilins. The absence of the covalent bond results in an extension of the π-system and a redshift of the absorption[5,19–21]. However, 27% of the ApcD3 sequences (to date) retain the bilin-binding Cys[78] residue[10,19], suggesting a degree of species specificity in the excitation energy transfer pathways to the photosystems via ApcD3.

A previously determined CryoEM map of a disconnected portion of the FR-APC core complex from *Synechococcus* PCC 7335 contained only three of the four expected phycobiliprotein trimers[10]. In the present work, we attempted the purification of a FR-APC + FR-PSII complex and obtained a structure of an intact bicylindrical FR-APC at a resolution of 2.51 Å, with a low-resolution ESP density localized where PSII is expected to bind (Supplementary Fig. 1). We describe the structure of the intact FR-APC complex containing all of the expected far-red paralogues and including an additional far-red specific subunit ApcF2. Moreover, an analysis of the planarity of bilin pigments and of their chemical environment in relationship to their absorption is provided, and suggests specific evolutionary traits needed to optimize far-red light excitation energy transfer.

## Results
### CryoEM data processing and map description
The CryoEM map of FR-APC was refined with C2 symmetry at a Gold Standard Fourier Shell Correlation (GS-FSC) resolution of 2.51 Å (Supplementary Fig. 1, Supplementary Fig. 2, Table 1). The map presents density to fit two antiparallel far-red light adapted allophycocyanin cores at an angle of ~30° to each other (see Methods section) (Fig. 1A). Each of the cylindrical allophycocyanin cores contains four (αβ)₃ trimers, arranged in two face-to-face [(αβ)₃]₂ hexamers.

The 1st and 2nd (αβ)₃ trimers, which are proximal to the thylakoid membrane plane, contain the far-red light specific subunits ApcB2, ApcD5, ApcD3, ApcE2, ApcD2, and ApcF2 as well as the white light shared subunit ApcC (Fig. 1D). As suggested previously[10], the second hexamer (formed by the 3rd and 4th (αβ)₃ trimers) is held in position by the second REP domain (REPeat domain) of ApcE2 and contains 6 ApcD5 and 6 ApcB2 subunits (Fig. 1D), but contrary to expectations[10], the distal ApcC subunit is not present. Due to its location, between 3rd and 4th (αβ)₃ trimers, the absence of the distal ApcC could be due to its loss during isolation, but the absence of any ESP density that could indicate partial occupancy, together with far-red light specific changes to the binding pocket, suggests that this subunit is natively absent from FR-APC cylinders. Even though some of the subunits (i.e., ApcB2 and ApcD5) are present in multiple copies in the structure, each bilin binding pocket is unique due to its position in the complex and the subunits that interact with it, independently fine-tuning their chemical environment and consequently their absorption spectra. This can tune specific sites and adjust the energy landscape of the pigments so that excitation can be efficiently funneled to the terminal emitter.

In addition, the map presents ESP density extending from the PB-loop of ApcE2 that can be attributed to the presence of a subpopulation of particles with connected photosystems. Unfortunately, the limited number of particles available, the low percentage of connected complexes and heterogeneity prevent further local refinement of the photosystem portion of the map. Attempts at resolving this density with local refinement and 3D classification yielded poor results (Supplementary Fig. 3).

### Absorption and low-temperature fluorescence spectroscopy
The absorption spectrum of the FR-APC complex presents the same features previously reported in literature, with two main absorption peaks at 650m and 710 nm, corresponding to the β-subunits and α-subunits, respectively[5,11,18]. Moreover, the absorption spectrum presents, although with lower intensity, a peak around 440 nm, that indicates the presence of chlorophylls from photosynthetic complexes in the sample (Fig. 2A). Further analysis with low temperature fluorescence emission spectroscopy indicates that when chlorophylls are excited with blue light (440 nm), there is a clear emission peak at 750 nm, with a shoulder at 730 nm, which can be interpreted as the fluorescence emission of FR-PSII, with some excitation being transferred back to the terminal emitter of the FR-APC. When exciting the sample with 550–600 nm light, where the absorbance of the FR-APC dominates, the fluorescence emission arises mainly from the bicylindrical FR-APC at 730 nm, with a smaller emission shoulder at 750 nm (Fig. 2B). This indicates that the sample is constituted mostly of disconnected FR-APC, with only a small proportion of the complexes being connected to FR-PSII.

### Coordination of bilin sites
The planarity and chemical environment of the bilin pigments strongly influence their absorption spectra[22,23]. For this reason, to gain insight into the structural origin of the red-shifted absorbance of some of the subunits of the FR-APC complex (Fig. 2A), the angles between rings A and B of each bilin molecule and those between rings C and D were measured (Fig. 3). The angles between rings B and C do not vary significantly and are therefore not included in the analysis.

The bilins contained in the α- and β-subunits of WL-APC[24] present very similar conformational landscapes, with ring D generally being between ~30°–~40° off plane and ring A presenting a larger spread of ~20°–~50° off plane (Fig. 3B). The bilins contained in FR-APC present a more diverse arrangement, with the β-subunits (ApcB2) maintaining similar torsion angles compared with WL in ring D, while the α-subunits (i.e., ApcD2, ApcD3, and ApcD5) are far more planar than their WL counterparts (Fig. 3C). The effect of this change can be seen in the absorption spectra of the FR-APC complex (Fig. 2A), where the two main absorption peaks can be attributed to the less planar β-subunits ( ~ 655 nm peak) and to the more planar α-subunits ( ~ 710 nm peak)[10,18]. The only exception is the bilin pigment in the β-subunit ApcF2, that presents an almost planar ring D compared to the ApcB2 subunits.

Previous structures reported the loss of cysteine linkage in ApcE2 (Fig. 4B) and ApcD3 (Fig. 5D)[10]. In comparison, the FR-APC complex from *C. thermalis* PCC 7203 also presents the cysteine-lacking β-subunit ApcF2. Comparing the chemical environments of the bilins present in the far-red specific subunits ApcE2 and ApcF2 with their white light counterparts ApcE and ApcF provides insight into the features that tune their site energy. The bilin present in the far-red specific membrane linker protein ApcE2 retains the same ZZZ,ssa conformation as in WL-ApcE (Supplementary Fig. 4). Nonetheless, the cysteine linkage is lost, extending the π-system of the pigment and allowing the bilin rings to be more coplanar (Fig. 4A–C), contributing to the significant redshift of the pigment[10,18].

In ApcF2, the linking cysteine is also lost compared to its white light counterpart, causing a change in configuration of the bilin pigment. While in ApcF the bilin is in ZZZ,asa, in ApcF2 the pigment is in ZZZ,ssa conformation (i.e. as in ApcE and ApcE2) (Supplementary Fig. 4). The flip of ring A in ApcF2 leads to the loss of the hydrogen bond between the methylated Asn[71] and the ring A nitrogen. These changes contribute to the red-shift from 651 nm in ApcF to 675 nm in ApcF2 in heterologously expressed proteins[14,25].

**Table 1 | Cryo-EM data collection, refinement, and validation statistics**

| Data collection | |
|---|---|
| Microscope | Krios III |
| Camera | Falcon 4i |
| Magnification | 155.000x |
| Voltage (kV) | 300 |
| Electron exposure (e-/Å²) | 40 |
| Defocus range (μm) | −0.8 to −2 |
| Pixel size (Å) | 0.723 |
| Energy filter | Selectris (20 eV) |
| Exposures | 9397 |
| Image format | EER |
| **Data processing** | |
| Box size | 700 px |
| Initial particles (no.) | 423,204 |
| Final particles (no.) | 36300 |
| Symmetry | C2 |
| Map resolution (Å) | 2.51 |
| Map sharpening B factor | −41.9 |
| **Model refinement** | |
| Refinement package | PHENIX |
| Initial model used | 8EQM |
| Real/reciprocal space | Real Space |
| Resolution cutoff | 2.60 |
| **Model validation** | |
| MolProbity score | 1.29 |
| ClashScore | 5.35 |
| Bond length R.M.S.D. (Å) | 0.002 |
| Bond angles (°) | 0.555 |
| Poor rotamers | 0.24% |
| Favored rotamers | 93.48% |
| Ramachandran outliers | 0.05% |
| Ramachandran favored | 98.67% |

The presence of two distinct non-covalently bonded ZZZ,ssa bilins allows us to compare their protein environments and suggest potential changes that further red-shift the ApcE2 pigment to suit it for its role as the putative terminal emitter. The chemical environments of the two chromophores differ in the following features. Firstly, ApcE2 features Trp[177] that sterically interacts with ring D, holding it in a more planar position and potentially π-stacking with it. In an equivalent position, ApcF2 presents Tyr[88] that does not interact with the chromophore. Secondly, in ApcE2 the keto group of ring D is H-bonded by the conserved ApcF/ApcF2 Thr[76]. Finally, the interaction partners of the propionate moieties are different. In ApcF2, Arg[77], and Arg[84] interact with both propionates of the bilin chromophore, whereas only the propionate of ring C interacts with ApcE2 Arg[120]. This could tune the electrostatic environment of the pigments, influencing their absorption properties.

To identify the amino acid residues responsible for red-shifting the absorption spectrum of α-subunits, far-red protein sequences were compared to their white light paralogues. ApcD5, ApcD3, and ApcD2 were analyzed in relation to ApcA rather than their paralogue ApcD1 since they occupy the same position structurally in APC cores[10,12,13]. The phycocyanobilins in ApcD5, ApcD3, and ApcD2, when compared to that in the white light expressed ApcA, have their rings A and D shifted to a more planar conformation with rings B and C. The bilin binding sites are located next to a protein loop which is different between far-red and white light paralogues (Fig. 5). A loss of this loop (residues AYG in ApcA) in ApcD5 results in a change in packing that influences the position of ring C and the hydrogen bond network of ring A (Fig. 5). The deletion displaces Asn[71] from the nitrogen of ring A, removing the hydrogen bond. In turn, the conserved far-red change in ApcD5, ApcD3, ApcD2 Thr[65] (Ile in ApcA) provides a hydrogen bond to the keto group of ring A. Moreover, the FR-specific ApcD5, ApcD3, ApcD2 Ile[118] (Thr in ApcA) also contributes to the shift of ring A by steric hindrance with its bulkier side chain.

The shift of ring D between ApcA and ApcD5, ApcD3, and ApcD2 could be attributed to the Tyr[84] to Trp[84] conserved change (Fig. 5A, B), resulting in a 0–~30° degrees distortion of ring D relative to ring C (Fig. 3B, Fig. 5B). The position of ring D in ApcD5, ApcD3, and ApcD2 is also influenced by a conserved Thr[75] to Cys[75] change in ApcB2. The bulky sulfur side chain of ApcB2 Cys[75] shifts ring D of the bilin in ApcD3, making it more coplanar with ring C. Finally, the binding of the propionate moiety of ring B also differs among the subunits. In ApcA, the propionate is not directly bound and is in proximity to an electron-rich residue (Met[77]). In ApcD5 and ApcD3, the propionate forms an ionic bond with Lys[74], while in ApcD2 it forms a hydrogen bond with Tyr[74]. This change could perturb the electrostatic environment of the pockets, influencing the absorption spectra of the pigments[26].

Some more changes differentiate the three ApcD paralogues present in FR-APC. Concerning ApcD3, one of the previously proposed emitters[10,11], Cys[75] is substituted in a large majority of the sequences with Ala[10]. Concerning ApcD2, the side chain of the Trp[84] FR-conserved change is in a different rotameric conformation. While in ApcD3 and ApcD5, this residue is perpendicular to ring D and sterically restraining it, in ApcD2 Trp[84] is parallel to ring D. As in the case of ApcE and ApcE2, this residue could π-stack ring D, influencing the electronic distribution of the bilin pigment and consequently its absorption[26]. No major structural changes are observed in the bilin chemical environment of the far-red adapted β-subunit ApcB2. Spectroscopically, the absorption spectrum of this subunit is very similar to that of the WL-paralogue ApcB[11,18].

As pointed out in the structural overview of the complex, the distal ApcC subunit, which in WL-APC is present between the third and fourth ring of the complex[24], is absent in this map of FR-APC (Fig. 6). In ApcE2 sequences, the loop interacting with the distal ApcC presents poor sequence conservation and some of the residues with bulkier side chains could create steric hindrance that would prevent ApcC from inserting in the distal site when compared with the white light paralogue ApcE (Supplementary Fig. 5). Moreover, even at high sigma values, no ESP density is present at the position normally occupied by ApcC, arguing against partial occupancy of this subunit.

Together these observations leave open the possibility that the absence of ApcC might be native rather than a purification artifact. By comparing the two relevant ApcB2 subunits, one in the first ring, which interacts with ApcC both in WL-APC and FR-APC (Fig. 6A) and one in the fourth ring that interacts with ApcC in WL-APC but not in FR-APC (Fig. 6B), it is clear that ApcC influences the conformation of bilin pigments.

The presence of the hydrogen bonding residue ApcC Gln[27], together with the steric effect of ApcC Phe[28], ensures that the orientation of ring D is fixed and in a more planar conformation when compared to the ApcB2 bilin in the absence of ApcC (Fig. 6), likely affecting the absorbance spectrum. The absence of the distal ApcC may tune the pigments in the distal fourth (αβ)₃ trimer, creating an energy gradient that facilitates excitation energy transfer to the terminal emitter ApcE2.

Given that the loss of cysteine linkage in ApcD3 is only found in a few species[11] and that FR-PSII subunits are well conserved amongst each other, it is likely that the excitation injection point of FR-PSII from FR-APC is also conserved. This therefore suggests that ApcE2 is likely the main terminal emitter, also consistent with the bilin in ApcE2 being the most planar (Fig. 3B) and forming an (αβ) heterodimer with the most redshifted β-subunit ApcF2. This, however, does not exclude potential contributions by ApcD3 or ApcF2, and a degree of species specificity in the excitation energy

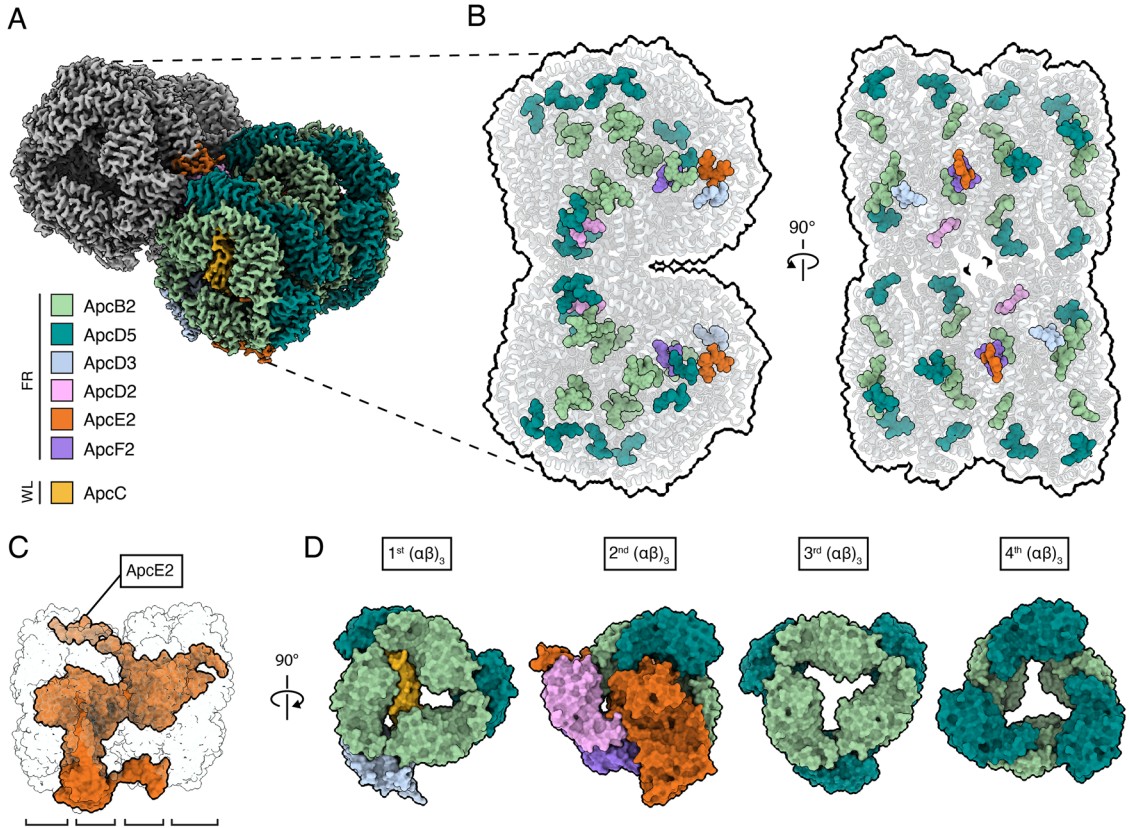

**Fig. 1 | Structure of the bicylindrical FR-APC and positions of the bilin pigments.**
**A** View of the dimeric FR-APC map from a tilted perspective facing the 1st (αβ)₃ from the cytoplasmic side. The map presents clear ESP for all the known FR-APC subunits, color key at the bottom left of the panel. **B** Location of the phycocyanobilin pigments in the bicylindrical FR-APC complex as viewed from the membrane plane and from the cytosolic side of the membrane. The pigments are also colored according to subunit coordination in (**A**). **C** Side view of a single FR-APC cylinder with ApcE2 colored in orange and every other subunit transparent. **D** Separate representation of the four (αβ)₃ trimers, also colored according to the color key in (**A**). It is important to note that the ApcE2 subunit is only represented in the second trimer.

transfer pathways to the photosystems in different far-red adapted strains. ApcE2 has also been suggested to have a key role in chlorophyll *d* synthesis[29]. However, in this structure, there are no other significant changes that would indicate an additional site for the synthesis of chlorophyll *d* by the FR-APC directly.

The dimer interface of the two cylinders is consistent with that predicted from the previous structural data of the monomeric first three rings of the complex[10], with ApcD2 being one of the main contributors by interacting extensively with an ApcB2 and an ApcD5 subunit of the third ring, and an ApcD5 subunit of the fourth ring in the neighboring cylinder (Supplementary Fig. 6B). Nonetheless, contributions to the structural integrity of the bicylindrical complex are not limited to this subunit. The *C*-terminal tail of the membrane linker protein, ApcE2, interacts extensively with an ApcD5 subunit in the third ring of the opposing cylinder (Supplementary Fig. 6A). Moreover, the ApcF2 subunit may interact with the ApcF2 in the opposing cylinder. When the two cylinders are brought in proximity by the dimerization, the two ApcF2 subunits are in position for long polar side chains to interact and provide stability to the complex. The high degree of flexibility of these sidechains, together with the poorer ESP density at the periphery of the map, does not allow these interactions to be observed.

## Discussion

This work provides the intact structure of the FR-APC bicylindrical antenna complex, offering an opportunity to gain insight into the conserved features that contribute to changes in optical properties in comparison with WL-PBS and the helical allophycocyanin (ApcD4/ApcB3) expressed during low-light

photoacclimation. Comparing the bilin pigments from subunits that are known to be involved in excitation energy transfer to the photosystems from WL-PBS, the bilin pigments in ApcE2 and ApcF2 are non-covalently bound and present a ZZZ,ssa configuration (Fig. 4), while ApcD3 is non-covalently bound only in a portion of the available sequences and presents a ZZZ,asa configuration (Fig. 5D). Nonetheless, while the ApcE2 bilin is almost completely planar, ApcD3 and in ApcF2 bilins present a decreased planarity of ring A, suggesting a relative blue-shift compared with ApcE2.

The four red-shifted ApcD paralogues (i.e., ApcD2, ApcD3, ApcD4, and ApcD5) all present a series of conserved features when compared with ApcA and ApcD1 that increase their planarity (Fig. 5). Conserved residues result in changes in the binding of ring A of the propionate moiety of ring B and of the planarity of ring D due to the presence of the Trp[84]. The only notable difference in the bilin binding sites between the α-subunits of FR-APC and helical-APC is the presence of the conserved Cys[75] in ApcD2, ApcD3, ApcD5/ApcB2 (Fig. 4), while in ApcD4/ApcB3 the typical ApcB Thr[75] is present. Nonetheless, while the two complexes have similar absorption profiles, the fluorescence emission spectrum of FR-APC is more redshifted (730 nm) (Fig. 2) compared to that of the helical-APC (718 nm)[18,30]. This difference is consistent with the identity of the excitation energy acceptor, with Chl *f* serving as the primary acceptor in FR-APC, while helical-APC likely transfers excitation to red forms of Chl *a*.

The structural basis for the observed blue shift and increased Stokes shift in the fluorescence emission of ApcD3/ApcB2 compared to that of other far-red-absorbing (αβ) heterodimers remains unclear. Heterologously expressed ApcD3/ApcB2 monomers exhibit an absorption maximum at 701 nm and an emission maximum at 734 nm, whereas

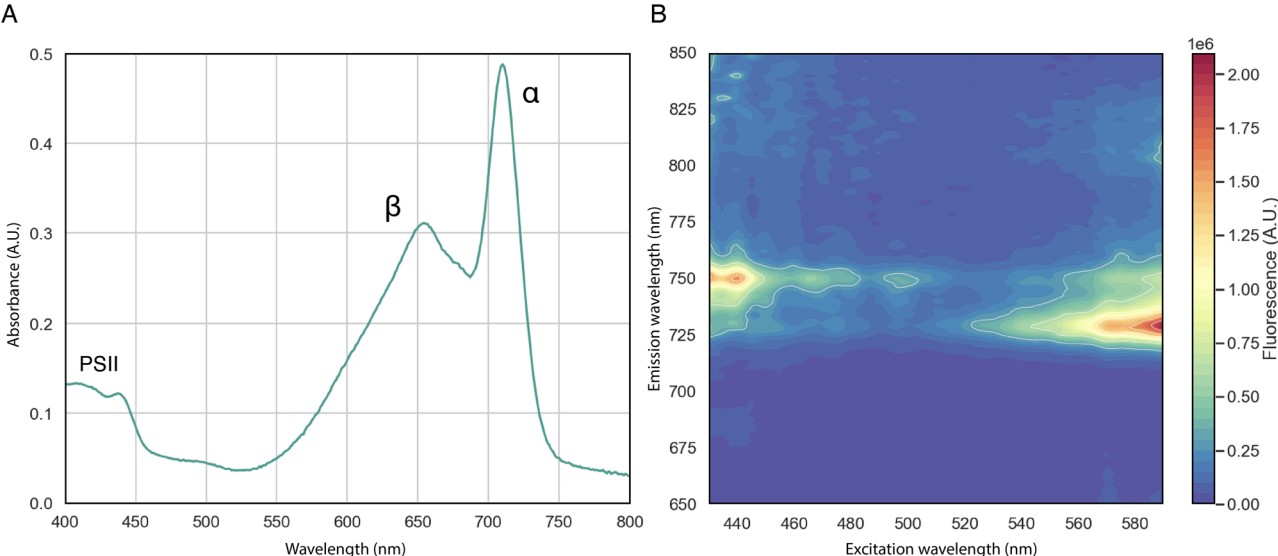

**Fig. 2 | Absorption and fluorescence spectra of the bicylindrical FR-APC complex. A** Room temperature absorption spectra of the bicylindrical FR-APC, Absorbance peaks corresponding to PSII chlorophylls, $\beta$-subunits, and $\alpha$-subunits are indicated. **B** 2D low-temperature fluorescence (77 K) excitation *vs.* emission spectra of the bicylindrical FR-APC. The intensity of fluorescence is represented by the color gradient.

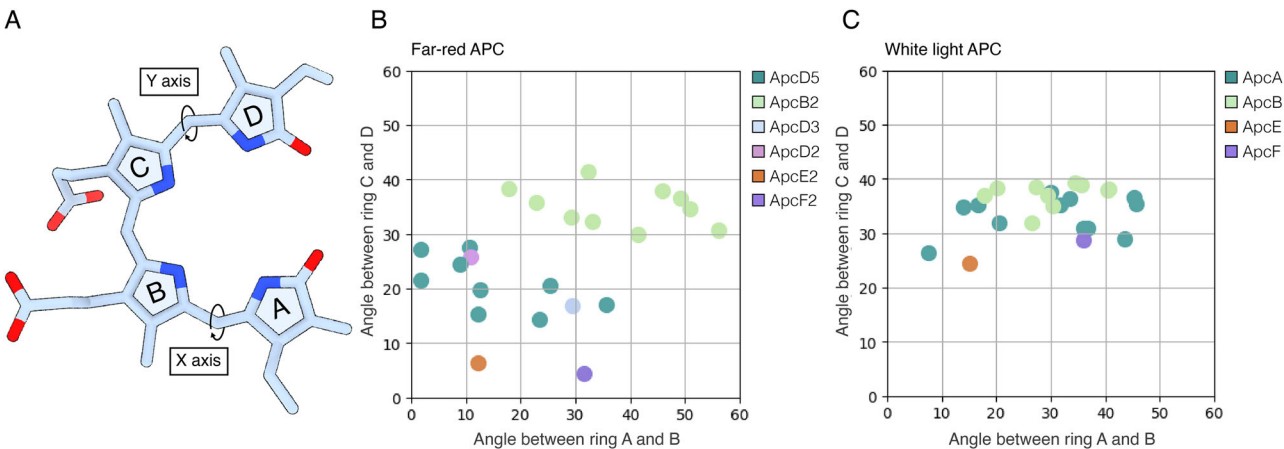

**Fig. 3 | Analysis of the planarity of the bilin pigments in FR-APC and WL-APC. A** Representation of an unbound planar phycocyanobilin pigment with the measured torsion angles indicated. **B** Variation of bilin planarity in FR-APC. The *x*-axis reports the degrees off plane of ring A with respect to ring B, while the *y*-axis reports the degrees off plane of ring D with respect to ring C. **C** Variation of the planarity of the bilin pigments in WL-APC from *Synechocystis sp*. PCC 6803 (7SC7). The *x*-axis reports the degrees off plane of ring A with respect to ring B, while the *y*-axis reports the degrees off plane of ring D with respect to ring C.

ApcD5/ApcB2 absorbs at 707 nm and emits at 712 nm[18]. Although these sites exhibit minor differences in sequence and bilin planarity, their distinct spectral properties cannot be fully explained based on current structural data. Nonetheless, the difference in fluorescence emission profiles between ApcD3/ApcB2 and the intact FR-APC complex suggests that ApcE2 is the primary terminal emitter. The fluorescence of the ApcD3/ApcB2 monomer is much broader, whereas the emission from the intact complex is sharper and slightly blue-shifted (Fig. 2B). This indicates that while excitation energy may localize at both ApcD3 and ApcE2, it seems that the latter contributes more significantly to the final emission[18]. Furthermore, the strong blue shift of ApcD3, with minimal spectral overlap with the fluorescence emission of the intact complex, suggests that during equilibration, excitation energy transfer is favored from ApcD3 to ApcE2 in comparison with the reverse direction.

The analysis performed here on the planarity of the bilin chromophores confirms that, as expected, ApcB2 contains the least planar pigments, together with the fewest amino acid differences, which likely explains why its absorption profile is more similar to its WL paralogue[13,18].

The structure of the bicylindrical FR-APC prompts questions on the importance of antenna size in FaRLiP. Phycobilisomes present a range of morphologies and evolutionary variability both in terms of absorbance profile and size, but to date, the smallest phycobiliprotein antenna system to be characterized is that found in far-red light acclimated organisms[12]. The FR-APC bicylindrical complex provides an extensive enhancement of the absorption cross-section in the far-red for FR-PSII, providing 24 far-red absorbing bilin pigments compared to the 10 long-wavelength chlorophyll pigments found in a FR-PSII dimer[6]. Nonetheless, as previously pointed out, the overall number of bilin pigments in FR-APC is much lower than in phycobilisome structures found in cyanobacteria grown in white light, where hundreds of bilins are present[12].

Why would a photosynthetic organism shaded from visible light sacrifice most of its light-harvesting capacity? The presence of smaller phycobilisomes allows cells to increase the thylakoid and thus photosystem density, an observation supported by the reduced distance between thylakoid membranes during FaRLiP[31,32]. This evolutionary adaptation might indicate that space optimization within and between

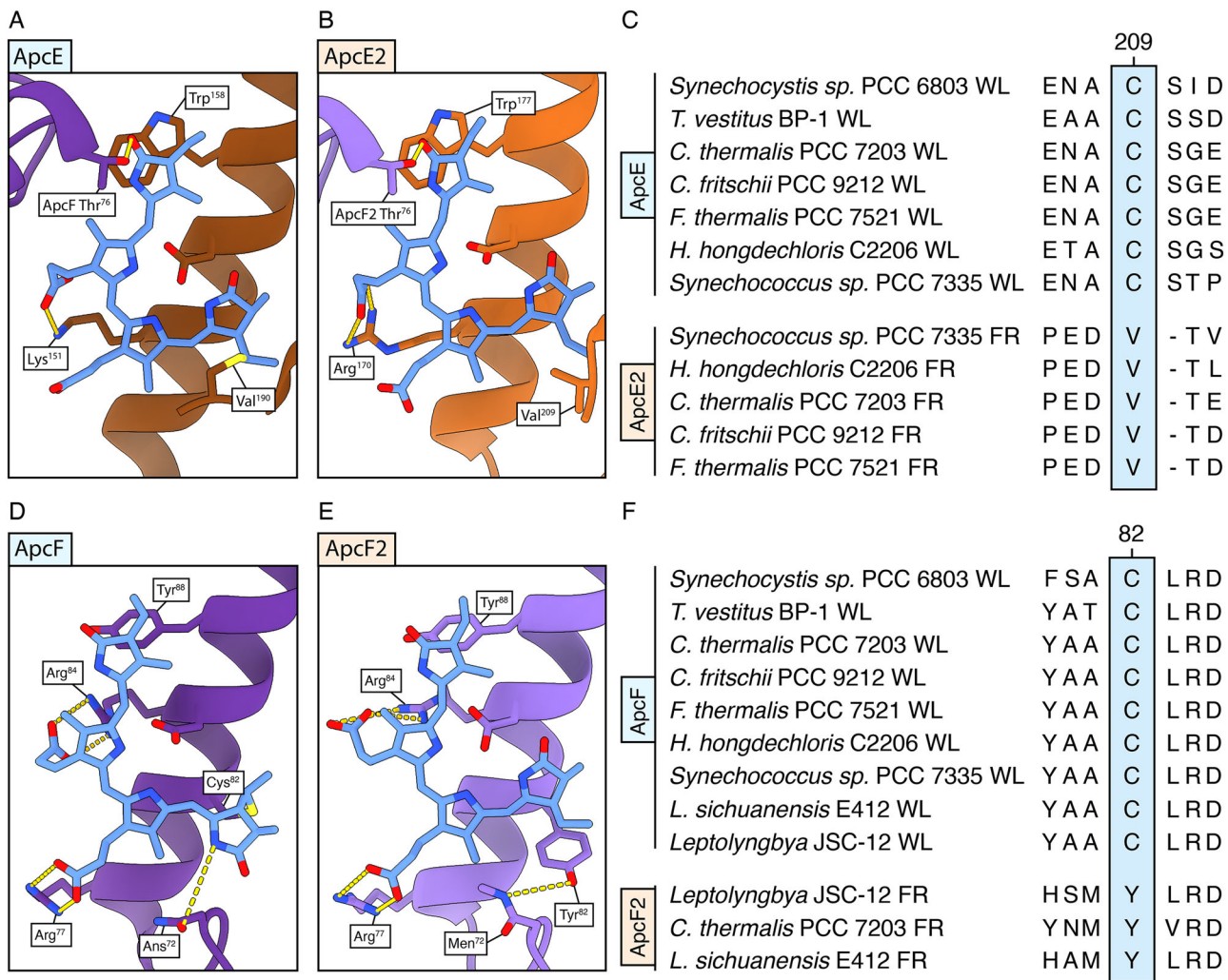

**Fig. 4 | Protein environment of the bilin in the ApcE2 and ApcF2 subunits.**
**A** Chemical environment of the bilin pigment in the ApcE protein (7SC7).
**B** Chemical environment of the bilin pigment in the ApcE2 protein. **C** Sequence alignment between selected ApcE and ApcE2 sequences. The light blue box highlights the discussed conserved loss of Cys between WL and FR paralogues.
**D** Chemical environment of the bilin pigment in the ApcF protein (7SC7).

**E** Chemical environment of the bilin pigment in the ApcF2 protein. **F** Sequence alignment between selected ApcF and ApcF2 sequences. The phycocyanobilin pigments are represented in blue, ApcE in brown, ApcE2 in orange, ApcF in purple, ApcF2 in violet, and hydrogen bonds in yellow. The light blue boxes in the sequence alignments highlight the conserved loss of Cys in the FR paralogues.

the membranes is preferred over antenna size in far-red light-adapted thylakoids. This reorganization could be advantageous by increasing the Chl *f*-to-bilin ratio, thereby enhancing absorbance at wavelengths beyond 710 nm.

Moreover, the differences in absorption spectra between far-red α- and β-subunits imply that the spectral range covered by the FR-APC evolved to be broader than the WL-PBS complex, even though WL-PBS are bigger and structurally more complex. The steeper absorption gradient of the bilins in FR-APC relative to WL-APC may result in a more efficient excitation transfer from FR-APC to FR-PSII, consistent with the enhanced trapping rate of FR-PSII when the FR-APC is connected[33].

The alterations during FaRLiP are not limited to the protein and pigment compositions of antenna proteins and photosystems. The ultra-structural changes in thylakoid organization during FaRLiP reflect the need in these conditions to maximize photosynthetic productivity with the decreased number of photons available. To gain insight into the distances between bilin pigments in the bicylindrical FR-APC and Chl *f* molecules present in FR-PSII, the structural models of FR-APC and FR-PSII[34] were fitted to previous CryoEM in situ single particle analysis structure of the PBS-PSII supercomplex from the cyanobacterium *Arthrospira platensis*[35] (Fig. 7A).

In the model obtained, the C2 symmetry planes of FR-PSII and bicy-lindrical FR-APC are not parallel but at an angle of ~20°, suggesting that the two cylinders of the FR-APC likely have two independent docking sites on FR-PSII dimers (Fig. 7B)[35]. This is confirmed when comparing the two opposite connection sites of the FR-APC cylinders to the array of FR-PSII complexes. On one side, ApcE2 is positioned directly above CP43, while on the other, ApcE2 is wedged between two adjacent FR-PSII dimers. The resolution of the *A. platensis* PBS-PSII supercomplex does not allow the identification of specific residues that stabilize these interactions, and dif-ferent species may have different interaction interfaces. For this reason, the model serves as a rough indication of distances rather than an actual representation of the array structures in far-red photoacclimated organisms.

Nevertheless, in this model, the closest assigned Chl *f* pigment in FR-PSII to the terminal emitter bilin in ApcE2 is C507. Chl *f* C507 is at an edge-to-edge distance of ~31 Å from the ApcE2 bilin directly above CP43 and is ~36 Å away from the ApcE2 bilin between the two dimers, consistent with previous structural and functional studies[10,33]. Moreover, the bilin pigment contained in ApcD3, previously proposed as a potential terminal emitter in FR-APC[10], is located further away from any assigned Chl *f* in FR-PSII when compared with the bilin in ApcE2, suggesting the role of the latter as the primary terminal emitter in FR-APC.

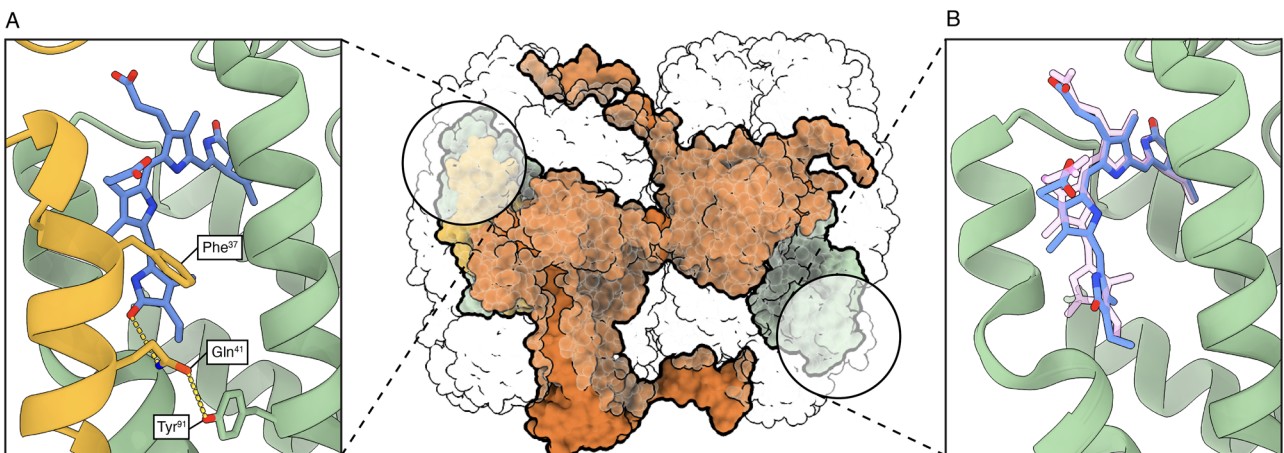

**Fig. 5 | Changes in the bilin sites of ApcD2, ApcD3, and ApcD5 compared with the WL paralogues.** Comparison of the chemical environment of the bilin pigment in the ApcA protein (**A**) against the ApcD5 (**B**), ApcD2 (**C**), and ApcD3 (**D**) proteins. The bilin pigments are always represented in blue. **E** Sequence alignment between ApcA and ApcD5, the light blue box highlights conserved changes in the sequence between white light and far-red paralogues, and the pink box highlights the loop deletion discussed in the text.

**Fig. 6 | Comparison between the chemical environments of different ApcB2 subunits in the FR-APC complex.** In the center, one cylinder with ApcE2 is represented in orange, ApcC is represented in yellow and two ApcB2 subunits in green. **A** Proximal ApcB2 bilin binding site in the presence of ApcC. ApcB2 is represented in green, ApcC is represented in yellow, and phycocyanobilin is represented in blue. **B** Distal ApcB2 bilin binding site in the absence of ApcC. ApcB2 is represented in green while phycocyanobilin is represented in blue. The position of the proximal bilin site is represented in transparent pink.

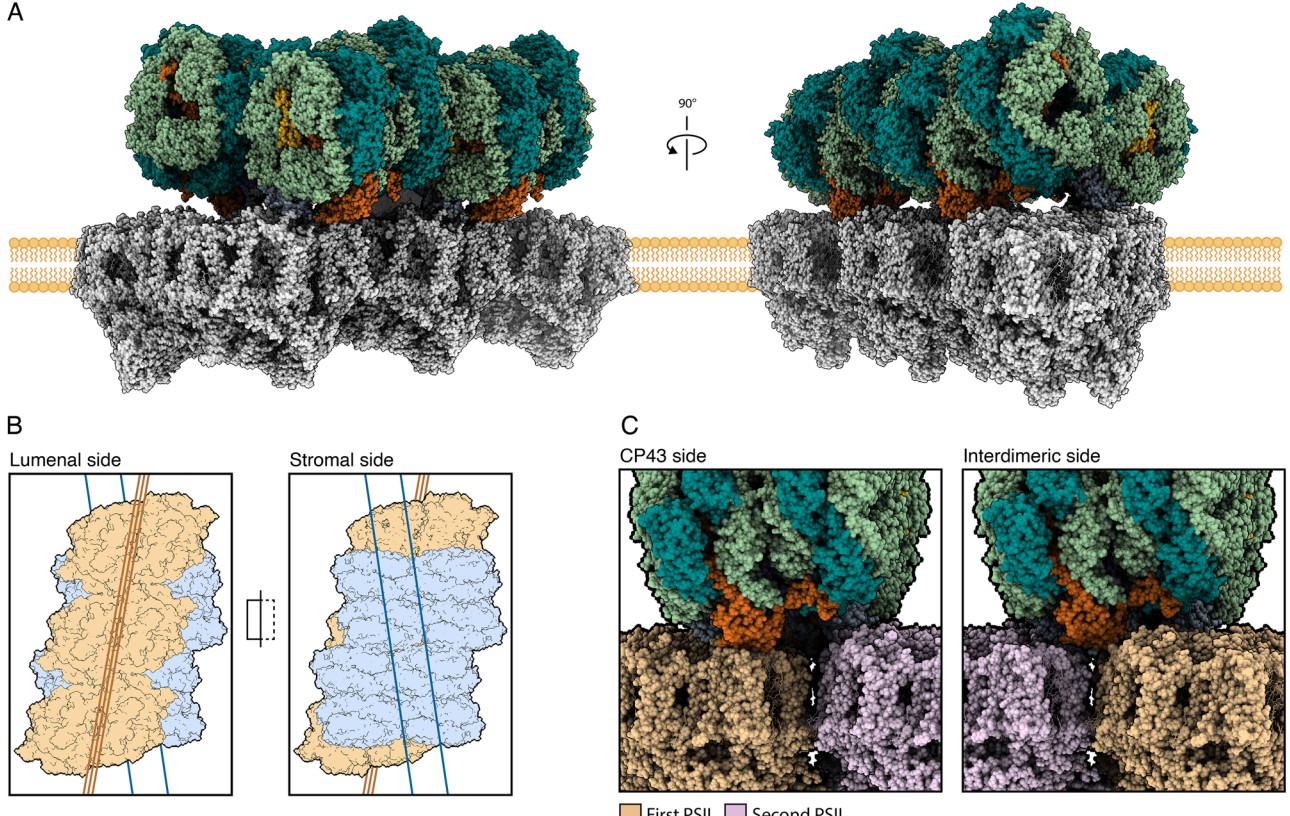

**Fig. 7 | Possible mesoscale arrangement of FR-PSII and bicylindrical FR-APC in thylakoid membranes. A** Representation of an array of three FR-PSII dimers connected to two bicylindrical FR-APC. The representation is modeled based on the arrangement of the PBS-PSII supercomplex from the cyanobacteria *A. platensis* (PDB 8WQL), FR-PSII from *Synechococcus sp.* PCC 7335 (PDB 8EQM) with the solved intact FR-APC. **B** Top and bottom view of the FR-APC + FR-PSII array complex with their respective C2 symmetry axes. FR-PSII is represented in orange, while FR-APC is in blue. **C** Close-up of the connection of FR-APC to FR-PSII on both sides of the FR-PSII array. In the left panel the interaction with CP43 can be seen and in the right panel ApcE2 is instead between two FR-PSII dimers on the opposite side. FR-APC is colored with the same color scheme as Fig. 1. The two adjacent FR-PSII dimers are represented in orange and pink.

While the docked model is not detailed enough to provide precise coordinates, it indicates that phycobilisome complexes, both in WL and FR organisms, are only capable of connecting to PSII when in an ordered array. In such an arrangement, it is possible that excitation energy transfer networks involve transfer between different bicylindrical FR-APC complexes, allowing excitation energy to be shared over the entire array of photosystems. Moreover, it is possible that the formation of the array itself is induced by the presence of ApcE2, stabilizing the side-by-side interaction of adjacent FR-PSII dimers.

From an experimental point of view, this behavior could explain the inability to resolve structures of APC-PSII complexes via CryoEM single particle analysis, as membrane solubilization treatments would likely destroy the supercomplexes and consequently cause the disconnection of the allophycocyanin cores from the photosystems. This emphasizes the need for alternative characterization strategies, with cryo-electron tomography being an obvious approach[35].

This work highlights the importance of understanding Chl *f*-based photosynthesis at a scale that goes beyond individual complexes and their interactions, extending to the organization of the thylakoid membranes and in situ photochemistry. Investigating excitation energy transfer networks in the mesoscale will lead to a better understanding of the relationship between antenna size, photosynthetic efficiency, and photodamage when comparing different evolutionary adaptations to far-red light and other ecological niches.

Ultimately, this study illustrates the evolutionary trade-offs that allow efficient management of excitation energy transfer by far-red light-adapted antenna. The insights obtained should help with the assessment of the feasibility, requirements, and limitations in the development of applications of far-red photosynthesis.

## Methods

### Culture growth

*Chroococcidiopsis thermalis* PCC 7203 was grown in liquid BG11 medium[36] at 30 °C under far-red light LEDs (750 nm, Epitex; L750-01AU) at an intensity of ~30 μmol photons m$^{-2}$ s$^{-1}$.

### Isolation of far-red adapted APC + PSII complexes

All purification steps were conducted in the dark and at 4 °C. Cells were harvested and washed in 20 mM MES, pH 6.5 buffer containing protease inhibitors (SIGMAFAST protease inhibitor tablets). Cells were broken with two passages in a continuous flow cell disruptor (Constant Systems) at a pressure of 39 kPsi and then centrifuged at $1000 \times g$ to remove any unbroken material. The supernatant was subsequently centrifuged for 20 min at 40,000 rpm in a Ti45 rotor at 4 °C to pellet the membranes. Membranes were resuspended in low salt buffer A (20 mM MES, pH 6.5, 5 mM MgCl$_2$, 5 mM CaCl$_2$, 1.2 M betaine, 0.03% $\beta$-DM), adjusted to a chlorophyll concentration of 0.4 mg/ml and solubilized at 4 °C for 1 h by the addition of $\beta$-DM to a final concentration of 0.4% (w/v). Solubilized membranes were centrifuged for 30 min at 42,000 rpm at 4 °C in Ti45 rotor to remove unsolubilized material. Isolation of protein complexes was carried out with a modified protocol based on Kern et al.[37]. The solubilized supernatant was loaded onto a DEAE Toyopearl 650S (ID: 50 mm Length: 450 mm), washed with one column volume of buffer A and eluted with a 0%–100% linear gradient of buffer B (20 mM MES, pH 6.5, 5 mM MgCl$_2$,

5 mM CaCl$_2$, 0.5 M NaCl, 1.2 M betaine, 0.03% β-DM) over five column volumes. Elution fractions containing protein complexes were concentrated (100 kDa or 50 kDa Filters, Amicon), washed with buffer A and stored at 4 °C or −80 °C, depending on the use.

## Grid preparation

Gold quantifoil (2 µm hole size and 1 µm hole spacing) with a 300 mesh grids (Quantifoil Micro Tools GmbH) were glow-discharged for 30 s at 25 mA. APC + PSII sample (~ 4.2 mg/ml of protein) was applied for 3 s at a temperature of 4 °C and 100% humidity in the presence of only dim green light, and the grids were blotted and plunge-frozen in liquid ethane with a Vitrobot mark IV (Thermo Fisher Scientific).

## Data acquisition

The particles were imaged using a Krios 3 operated at 300 kV. Images were recorded on a Falcon 4i with a pixel size of 0.723 Å and an exposure of 40 electrons per Å$^2$ for a total of 40 frames. Images were collected in super resolution mode with a SelectrisX energy filter with a slit width of 20 eV. The targeted defocus range was varied from –0.6 to –2.4 µm using the EPU software (Thermo Fisher).

## Single particle analysis

A total of 9397 movies were collected. Frames were aligned, dose weighted, and the contrast transfer function (CTF) was estimated in CryoSPARC v4.0.2[38].

Micrographs were curated by removing ones with CTF fits below 10 Å. The subset obtained contained 9224 micrographs (98%) and blob picker was used to pick particles in a subset of micrographs across the defocus range. Particles were 2D classified and used to train Topaz[28] to pick particles across the entire dataset, yielding 423,203 picks. After multiple rounds of 2D classification and ab-initio refinement, duplicated particles were removed and a subset of 36,300 particles were used for homogeneous refinement to obtain an initial map >5 Å and to confirm C2 symmetry. After multiple rounds of per particle CTF refinement and local motion correction[39], a set of 36,300 particles was used to perform non-uniform refinement[40] imposing C2 symmetry, producing a map at a global resolution of 2.61 Å based on GS-FSC at a cut-off of 0.143.

## Model building

The FR-APC incomplete rod from 8UHE[10] was fitted to the ESP map using the Phenix software suite[41], and then the third ring was copied and fitted again with Phenix into the density of the fourth ring. The complete model was then mutated to the correct sequence with Chainsaw (CCP4) and refined in Coot[27]. The amino acid for each subunit's density was evaluated individually, and the model was then refined in real space in Phenix (Table 1).

## Absorbance measurements

Absorption spectra of isolated complexes were recorded at room temperature with a Cary 60 spectrophotometer (Agilent technologies) in buffer A.

## 77 K fluorescence spectroscopy

Low temperature fluorescence measurements were performed on isolated FR-PSII + FR-APC complexes at an OD$_{710}$ of 0.1 with a FluoroMax 4 spectrofluorometer (HORIBA Ltd., Japan). 77 K fluorescence emission was measured from 650–850 nm with excitation and emission slits of 2 nm and an integration time of 0.5 s using an excitation range from 400–600 nm.

## Bilin angles measurements

To assess the planarity of bilin chromophores, we measured the angles between planes defined by specific atom triplets within the bilin rings of CYC molecules in a protein structure. Ring A plane was defined by the NA, C3A, and C4A atoms, ring B plane was defined by the NB, C3B and C4B atoms, ring C plane was defined by the NC, C3C and C4C atoms and ring D

plane was defined by the ND, C3D, and C4D atoms. Plane normals were computed using the cross product of two vectors within each plane, and the angles between planes were calculated using the dot product and arc-cosine function. Angles were then adjusted to a centered range where values above 90° were transformed to 180° – angle. The method was applied to all CYC residues in the structures, and the resulting angles were stored in a data frame for further analysis.

## FR-APC cylinders twist measurement

The degree of twist between the two FR-APC cylinders was calculated as follows. The geometric center of each (αβ)$_3$ ring was calculated using only the backbone atoms and excluding ApcC and the REP domains of ApcE2. The cylinder axes were estimated as the line of best fit through the geometric center of the four (αβ)$_3$ rings. The angle between the vectors of the axes was calculated, which corresponds to the degree of twist between the two cylinders.

## Reporting summary

Further information on research design is available in the Nature Portfolio Reporting Summary linked to this article.

## Data availability

The atomic coordinates have been deposited in the PDB with accession code 9I1R and on EMDB with accession code EMD-52573. All other data are available from the corresponding author on reasonable request.

## Code availability

The scripts used in this work are available at https://github.com/giovanniconsoli/phycobiliprotein-tools.

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

## Acknowledgements

We thank Kenta Renard for the numerous discussions. We thank the center for structural biology at Imperial College for the training provided, for help with the early stages of sample screening and with data collection. We thank Diamond for the access and support of the cryo-EM facilities at the UK national electron Bio-Imaging Center (eBIC), proposal BI25127. This project made use the Computing Platform for Electron Microscopy at Imperial College funded by the BBSRC Mid-range equipment Initiative 22ALERT (BB/X019284/1). We thank the reviewers for the constructive discussion on the features that tune the bilin absorption spectra. Funding information, including grant numbers, complete funding agency names, and recipient's initials: Marie Skłodowska-Curie grant agreement, No. 955520 – GC, AWR, AF. Biotechnology and Biological Sciences Research Council, BB/R001383/1 – AWR, AF, GAD. Biotechnology and Biological Sciences Research Council, BB/V002015/1 – AWR, AF, JWM. Biotechnology and Biological Sciences Research Council, BB/R00921X – AWR, AF, GAD. Leverhulme Trust Grant RPG-2022-203 – AWR, GAD, AF. The Royal Society (Royal Society Research Professorship 2024) – AWR. Biotechnology and Biological Sciences Research Council, BB/Z516740/1 – GC, AWR, AF.

## Author contributions

G.C. and H.F.L. initiated the study. G.A.D. grew and harvested *C. thermalis* PCC 7203. G.C., H.F.L., G.A.D., A.F., and A.W.R. conceived of the main experiments, collated results and interpretations, and wrote the article with input, edits, and approval from all authors. G.A.D. developed and performed the isolation of the complexes. G.C. and H.F. prepared the grids. G.C. and H.F.L. collected the micrographs. G.C. and H.F.L. processed the data to obtain the map. T.R., A.M., G.C., H.F.L., and J.W.M. built the atomic model. HFL collected the sequences and performed the phylogenetic analysis. G.C. coded the bilin planarity quantification method. G.C. prepared the figures with input, edits, and approval from all other authors.

## Competing interests

The authors declare no competing interests.
