## [Transparent Peer Review file · Communications Biology]

Structure of a stripped-down and tuned-up far-red phycobilisome

Corresponding Author: Professor A Rutherford

Version 0:

Reviewer comments:

Reviewer #1

(Remarks to the Author)

Summary

The manuscript "Structure of far-red allophycocyanin: stripped down and tuned up for low energy photosynthesis" by Consoli, Leong, Davis and co-workers reveals the structure of the far-red light bicylindrical allophycocyanin complex (FR-APC) from the cyanobacterium *Chroococcidiopsis thermalis* PCC 7203. Key insights include the fourth ($\alpha\beta$)₃ trimer, which was missing in a previous FR-APC structure from a different organism, as well as, importantly, the assembly of two C2 symmetric bicylindrical cores. This assembly closely matches the assembly of APC in white-light phycobilisomes (WL-PBS), further substantiating the hypothesized models of the APC--photosystem II (PSII) interface. This is an important insight for modeling the energy transfer of far-red light into FRL-PSII. However, there are numerous aspects of the manuscript that require improved analysis and additional data.

While I find the paper interesting for expanding on previous work, substantial improvements to the manuscript are necessary before publication. Below are major critiques followed by some minor suggestions.

Major Critiques

1. Page 4. Regarding the observed absence of ApcC in the 4th trimer, the claim that ApcC is "unlikely to be lost" should be substantiated by empirical data. In the WL-PBS structures, ApcC appears fairly solvent exposed, enough to conceivably disassociate. Are there sequence differences in the FR-APC subunits that suggest a weaker ApcC interaction? Can a docking measurement/MD simulation be used? Are there any biochemical or spectroscopic results which agree with this interpretation? Without empirical data, this conclusion seems speculative.
2. Page 4. Please elaborate on the attempts to resolve a structure of FR-APC in complex with FRL-PSII. While the authors claim that there is some signal (at extremely low contours) consistent with the PSII binding site, there are slim data in support of this. 2D classes with the FR-APC and FRL-PSII should be observed in the data, but none are shown in the SI, possibly a result of the rather small boxsize for such a complex. The included micrograph clearly shows PSII particles, so some PSII classes should be resolved. I suspect they were simply not included/mentioned. Please elaborate on the methods used to try to evaluate the FR-APC---FRL-PSII complex (e.g. local refinement, 3D classification, manual picking). Negative stain data should also provide insight into the assembly/disassembly, but there is no mention of negative stain data.
3. Page 6, "550 – 600 nm light, where the only species absorbing are the bilin pigments". Please phrase this more carefully, as PSII does have absorbance in those wavelengths.
4. Page 7. Please clarify how the torsion angle measurements were made. Which four atoms were used? Fig. 3A seems to indicate that four dihedral measurements were made, but the data show only two dihedrals. Furthermore, the measurements should all be provided as a SI table, including the measurements between ring B and C as a semi-control.
5. Page 7-8, "...given that the loss of cysteine linkage in ApcD3 is only found in a few species ... suggests that ApcE2 is the main terminal emitter...". Can this be supported by experimental data? If this is speculative, please clarify the wording accordingly.
6. Page 8, Fig. 4. "protein backbone in orange". Remove 'backbone'. Side chains are also shown.
7. Page 8, "when compared to the one in WL-ApcA". How valid is this comparison? Please clarify this for the reader. My understanding is that the present FR-APC is from a different organism from both the previous FR-APC and the WL-APC being compared against. Please be careful, here and throughout the manuscript, to clarify if there are any sequence differences that should be considered alongside any structural comparisons. Show accompanying justification and data in the SI.
8. Page 9, Fig 5. Panel E is mislabeled.

9. Page 9, the ApcB2-Cys75 and “loop between two helices” are not indicated in Fig. 5 despite the reference.
10. Page 10 “... steric hindrance that would prevent ApcC from inserting...”. I echo my first critique that additional evidence is necessary to conclude that ApcC is natively excluded, not just lost in the present sample. I am not fully convinced that the interactions shown in Fig. S2 are substantial enough to prevent the entire ApcC subunit from binding. By aligning the WL APC structure, the ApcC subunit does not appear to globally clash with FRL-ApcE2. On the contrary, many of the polar-polar interactions appear conserved. The clashes described in Fig. S2 could conceivably be accommodated by a local backbone shift. Furthermore, I reiterate my previous comment about the comparison between WL-APC and FRL-APC, that the organism and possible sequence differences needs to be clarified and considered. Please expand on this analysis, as it is a key point in the current manuscript.
11. Page 10, regarding the Gln40 and Gln41 interactions. Firstly, the Gln40 and Gln41 conformations shown in Fig. 6 do not seem to match the conformation in the provided PDB. Please verify this point. Furthermore, the Gln conformations discussed are not the conformations shown in previous structures, both FR and WL. While it is true that the orientation of the amide’s carbonyl and amine groups are not easily differentiated structurally, the nearby ApcB2 Tyr87 seems to be a more likely proton acceptor for ApcC-Gln41 and a backbone carbonyl Asn110 seems to be a more likely proton acceptor for ApcC-Gln40. The bilin carbonyl seems a less likely proton acceptor for either of these Gln side chains. This is consistent with previous FR and WL structures, and the current FR-APC cryoEM data. Furthermore, the geometry to support the Gln-amine to bilin-carbonyl H-bonding is far from ideal for either Gln40 or Gln41, nor are they particularly supported by the cryo-EM data. Rather, consider that ApcC-Phe37 and ApcB-Tyr87 constrain the bilin ring.
12. Page 10 “bilin pigment is slightly stretched”. Please provide a measurement and report the difference.
13. Page 11, “interacts extensively with an ApcD5 subunit in the third ring...”. This point is not obvious in Fig. 1. In Fig S3, all the depicted interactions should have their side chains labeled. The H-bond distances should also be reported, either directly in the figure, or as a SI table. Some of the H-bonds, especially in panel C, appear too long. Overall, increased specificity of the data will strengthen the author’s claim.
14. Page 12 “which likely explains why its absorption profile is similar to its WL paralogue”. Please provide a reference for this.
15. Page 12, “This evolutionary adaptation indicates ...”. I am under the impression that the phycobilisome has several morphologies across different cyanobacterial species and that this is not limited to WL vs FRL/LowLight. Is it technically correct to say that the difference in morphology is directly caused by the evolutionary pressures of different photo-niches? Please expand on this point for the reader, providing references. I also note that this point is referenced in the Abstract and would likewise benefit from additional discussion of the literature.
16. Page 14, Fig. 7. Please emphasize for the reader that these models are based on the previous in situ results. Please also provide their PDB code in the figure description.
17. Page 14. “not parallel but at an angle of roughly ~20°”. Zhang et al. 2024 report a 6° difference. Please clarify this discrepancy. (Zhang et al. 2024 report a 25.3° twist of the APC cores, a measurement that should similarly be reported on the current FR-APC structure).
18. Page 17. “GC-Fourier Shell Correlation”, Typo. GS is correct, but “gold standard” is better.
19. Page 23. “script ... is available upon request”. Basic details about what the script does and how it is used should be described in the methods section. See critique #4.
20. Table S1. The final particle count does not match what is in the Methods section.
21. Table S2. Please review existing cryo-EM literature to see how these values should be reported and which values are needed. Clash score, for example, is missing from your table.
22. Fig. S1. The 2D classes are mislabeled.
23. Fig. S1. Please review recent cryo-EM literature and include standard data. Several data are missing, for example, the FSC curve (the cFSC is not a replacement, as it evaluates orientation bias), map-model FSC curve, angular distribution plot, a clearer local resolution map, etc. Furthermore, the workflow diagram should be much more substantial.
24. Minor grammatical errors are found throughout.

Minor Suggestions:

1. Page 8, Fig 4. Labeling ApcE as WL and ApcE2 as FR might improve readability.
2. Page 8, Fig 4. Throughout the manuscript and SI, coloring/formatting the sequence alignment using ClustalX formatting is more informative and more consistent with existing literature (especially outside the photosynthesis field).
3. SI, Fig. S3. In panel B, the intra-helix H-bonds are unnecessary and confuse the inter-subunit interactions being discussed.
4. Page 9, Fig 5. In panel A, ApcA-Asn71 appears to H-bond with the bilin from this angle, even though this is not the case. Asn71 can be hidden to avoid confusion. Furthermore, consider showing H-bonding using dashed lines.
5. Page 9, Fig 5. Panel A is improperly cropped.

Reviewer #2

(Remarks to the Author)

The MS from Consoli et al describes the cryoem structure of a bicylindrical APC induced by far red (FR) light from a *C. Thermalis*. This is a nice addition to the field which complements and improves the description of light harvesting systems in under FR conditions, a subject of considerable general interest. Compared to previous studies a more intact form of this APC was obtained, which contains the “missing” ab trimer and retains the overall bicylindrical organization. The photophysical properties of this sample are probably a better representation of the in vivo APC than previous structures. The presentation of the work is good, and the quality of experimental data is high. In general, I think the MS should be accepted with minor modifications only.

Minor points -

Line 73 contains "Click or tap here to enter text."

In the provided model Chain L/I Ala67 seems to be a different amino acid.

Fig. S1 – the use of C2 symmetry is mentioned in the text, but the symmetry used in refinement should be indicated at the appropriate steps in section A.

Overall clashscore and molprobrity score for the final model should be included in table s2.

Reviewer #3

(Remarks to the Author)

The manuscript by Consoli et al. reports a cryoEM structure at 2.6Å resolution of far-red allophycocyanin (FR-APC) in a bicyclic antenna complex. The contribution of this work to the field/topics of light-harvesting and photosynthesis in cyanobacteria can be summarized in two aspects. First, this is the first intact cryoEM structure reported for the FR-APC complex where all four expected phycobiliprotein trimers are present. The remarkable conformational diversity of PCB revealed by the FR-APC structure allows in-depth examination of the molecular basis underlying the PCB-based far-red absorption. Second, the authors presented a model of FR-APC docked onto FR-PSII using the published cryoET data as a guide. In the absence of an experimental structure, such a model provides an important structural glimpse into the directed energy transfer from the bilin pigments of APC to the chlorophyll f in PSII. Specifically, this docking model suggests that ApcE2 (not ApcD3) is the main terminal emitter in FR-APC. Overall, this manuscript is well written, and the findings are interesting. However, some weaknesses need to be addressed before publication. Since the bilin conformational diversity is the forte of this paper, it requires in-depth analysis of the protein-chromophore interactions beyond the planarity analysis (Fig. 3). Specifically, plain description of the protein environment is not sufficient for revealing the structural basis of far-red absorption of PCB. Given the excellent resolution and diverse configurations of PCB, this cryoEM structure offers a unique opportunity to identify key interactions (again beyond planarity) responsible for extending the absorption spectra of PCB to the red and far-red regions. See below for comments.

Major:

- Please provide the focused (masked) density maps (in supplemental materials) to justify the structural interpretation (Z/E and syn/anti) of all PCBs in the FR-APC complex.
- The FR-APC complex features PCB in diverse configurations (Fig. 4 and Fig. 5). Judging from the views provided, ApcE features PCB in ZZZ,ssa while PCB in ApcE2 adopts an all-Z,syn configuration (Fig. 4). Both ApcA and ApcD5 show PCB in ZZZ,ass while ApcD2 and ApcD3 show PCB in ZZZ,asa, which is typically found in phycobiliproteins (Fig. 5). Such diversity is highly interesting and needs better illustration. A multi-panel figure that shows the PCB structures (in a face-on view NOT side view) from different subunits of FR-APC (perhaps including WL-APC) would be helpful to highlight such diversity and facilitate direct comparisons.
- It is well known that other than the bilin planarity (depicted by the twist angles between pyrrole rings in Fig. 3), specific protein-PCB interactions play critical roles in spectral tuning. Please analyze and discuss what unique protein-chromophore interactions in the chromophore-binding pocket, including ionic interactions, geometric factors or presence of specific Cys anchors, contribute to the tuning of the absorption spectra, in particular, the far-red absorption. Plain description simply does not do the job. Further analysis with tabulation and plots (against the peak wavelengths) are needed. Discussion may also cover the current understanding of spectrum-tuning mechanisms obtained from other bilin-binding proteins, e.g. cyanobacteriochromes (CBCRs).
- The modeling of the APC-PSII complex is interesting and significant. Perhaps, these texts are better suited for the Results section. An alternative would be to combine Results and Discussion as one section.

Minor:

Please mark the C2 symmetry in Fig. 1B (right panel).

Line 248-251: the statement that "The bilin pigment in ApcE2 is non-covalently bound and almost completely planar, pushing the physical properties of phycocyanobilin towards the maximum red-shift attainable by this pigment (Staheli et al., 2021; Miao et al., 2016)." is not accurate. The maximum red-shift reported for PCB has exceeded 740nm for a far-red cyanobacteriochrome exhibiting an all,Z-syn conformation covalently anchored to a Cys residue (Rockwell et al, 2016).

Version 1:

Reviewer comments:

Reviewer #1

(Remarks to the Author)

The manuscript is much improved and all my comments have been adequately addressed. The additional analysis and discussion prompted by reviewer 3 is an excellent addition. I have a few minor points that should be addressed, but overall, the manuscript is in good shape.

1. Introduction final paragraph: "density localized where PSII is expected to bind." Some of the 2D classes shown in Fig. S1 are data in support of this point. This could be mentioned, and an enlarged figure of a 2D class can be shown in Fig. S6.
2. Results paragraph 1: The ~30° angle measurement is a good addition but please modify the wording to make it clearer for

a new reader.

3. I like the changes made to Fig. 4 and 5 based on reviewer 3's feedback, however the side view is still of interest, especially since portions of the discussion emphasize the planarity. I suggest the original figures be included as SI, perhaps in a single composite figure omitting the sequence alignment but showing both perspectives of the bilins.

4. Fig. 6. ApcC Gln40 may be a typo as ApcC Gln41 is closer to Tyr91. Furthermore, in the updated pdb file the ApcC Gln41 amide orientation appears flipped relative to what is shown in Fig. 6. The software can sometimes do this without a clear warning, so please confirm the orientation is as intended by the authors.

5. Right before Fig. 7, the first in situ seems to be redundant.

6. If the authors see fit, there may be some opportunity to improve the wording of the final few paragraphs of the Discussion, which are somewhat repetitive.

Reviewer #2

(Remarks to the Author)

The authors addressed all of my concerns and I support the publication of the manuscript.

Reviewer #3

(Remarks to the Author)

The revised manuscript has properly addressed my questions/concerns from the first round of review. However, I do see extensive responses to the reviewer #1's questions, which appear critical but fair. To this end, I would support the publication of the revised manuscript if reviewer #1 is satisfied with the responses from the authors.

Reviewer #1 (Remarks to the Author):

Summary

The manuscript “Structure of far-red allophycocyanin: stripped down and tuned up for low energy photosynthesis” by Consoli, Leong, Davis and co-workers reveals the structure of the far-red light bicylindrical allophycocyanin complex (FR-APC) from the cyanobacterium *Chroococcidiopsis thermalis* PCC 7203. Key insights include the fourth ($\alpha\beta$)₃ trimer, which was missing in a previous FR-APC structure from a different organism, as well as, importantly, the assembly of two C₂ symmetric bicylindrical cores. This assembly closely matches the assembly of APC in white-light phycobilisomes (WL-PBS), further substantiating the hypothesized models of the APC--photosystem II (PSII) interface. This is an important insight for modeling the energy transfer of far-red light into FRL-PSII. However, there are numerous aspects of the manuscript that require improved analysis and additional data.

While I find the paper interesting for expanding on previous work, substantial improvements to the manuscript are necessary before publication. Below are major critiques followed by some minor suggestions.

We thank the reviewer for the thorough comments and analysis of the manuscript. We have addressed all the major and minor points and we hope that the improved manuscript will be of the quality expected. We want to inform the reviewer that we have now obtained a slightly better resolved map (2.51 Å) of the complex and that we have redeposited it under the same accession code. Here is a link to the new maps and model:

https://drive.google.com/drive/folders/1GL9zWQVeEZr_DifUfGSyLxPhKA8OQLGR?usp=drive_link

Major Critiques

1. Page 4. Regarding the observed absence of ApcC in the 4th trimer, the claim that ApcC is “unlikely to be lost” should be substantiated by empirical data. In the WL-PBS structures, ApcC appears fairly solvent exposed, enough to conceivable disassociate. Are there sequence differences in the FR-APC subunits that suggest a weaker ApcC interaction? Can a docking measurement/MD simulation be used? Are there any biochemical or spectroscopic results which agree with this interpretation? Without empirical data, this conclusion seems speculative.

We appreciate the concerns presented for this point. Currently, biochemical or spectroscopic *in vivo* data that can confirm the physiological absence of ApcC from the interface between the 3rd and 4th cylinder of the FR-APC complex are not available.

Our initial statement didn't exclude the possibility that ApcC could spontaneously disassociate and was based on three main points. First, the low degree of conservation of the interaction area of ApcC with AcpE2, including conserved residues that contribute to the rigidity of AcpE2 and remove hydrogen bond contacts to ApcC. Second, the position of ApcC, even if solvent

exposed is between the 3rd and 4th ($\alpha\beta$)₃ trimer and it seems unlikely that the degree of motion of this area is enough to allow the dissociation of this protein. Third, if ApcC was lost during the isolation of the complex it would be possible to see the presence of an ESP density in the map at least at high sigma values due to low partial occupancy. The complete absence of any density was to us more likely to indicate that the distal ApcC is not present at all.

Nonetheless, to further investigate the two hypotheses we performed docking of the ApcC subunit, unfortunately obtaining inconclusive results. Clashes are still present in the docked model. We evaluated the possibility of performing MD simulation of the protein assembly, but the scale of the complex would require us to only investigate short timescales or to perform a coarse analysis, and neither of the two would be helpful in obtaining a definitive answer.

While we cannot rule out a loss of a distal ApcC, our structure from *Chroococidiopsis thermalis* and the previously reported structure from *Synechococcus sp.* PCC 7335 (Gisriel et al. 2024, *JBC*) both lack ApcC in this position. With both dataset structures, from two different organisms, the question must be asked whether there is an ApcC in this location and it is easily lost or whether there in fact is not an ApcC in this position in the FaRLiP APC. The currently available data cannot rule out either scenario, but what they current data seem to consistently show is that there is not an ApcC in the distal position. We have less data (at this time none) to suggest that it is lost during purification in either species, and therefore our best interpretation with the data at hand is to interpret the APC structure has not having this ApcC.

We have clarified and rephrased the main text to reflect the strength of our conclusions and to accommodate both scenarios (page 5, rows 14-17): “the absence of the distal ApcC could be due to its loss during isolation, but the absence of any ESP density that could indicate partial occupancy, together with far-red light specific changes to the binding pocket, suggests that this subunit is natively absent from FR-APC cylinders.”

2. Page 4. Please elaborate on the attempts to resolve a structure of FR-APC in complex with FRL-PSII. While the authors claim that there is some signal (at extremely low contours) consistent with the PSII binding site, there are slim data in support of this. 2D classes with the FR-APC and FRL-PSII should be observed in the data, but none are shown in the SI, possibly a result of the rather small boxsize for such a complex. The included micrograph clearly shows PSII particles, so some PSII classes should be resolved. I suspect they were simply not included/mentioned. Please elaborate on the methods used to try to evaluate the FR-APC---FRL-PSII complex (e.g. local refinement, 3D classification, manual picking). Negative stain data should also provide insight into the assembly/disassembly, but there is no mention of negative stain data.

Our claims that we had hoped to structurally resolve an intact FR APC-PSII came initially from the development of a native photosystem isolation procedure as described in the materials and methods. To avoid using sucrose gradient purification of the thylakoid membrane

complexes in order to obtain active PSII complexes. Not knowing the separation and elution behavior of these complexes in the DEAE-Toyopearl, all eluted fractions were initially tested for their ability to evolve oxygen. To our great surprise, a fraction comprising nearly negligible amounts of chlorophyll and that in fact blue in color evolved oxygen, which eluted far after any other FR-PSII complexes. While this did not prove that the sample was functionally connected FR-PSII+FR-APC, the small amount of sample available motivated us to attempt to resolve the structure(s) of the sample as there was not enough for a full spectroscopic analysis.

The suggested of at least a subset of the sample comprising connected FR-PSII+FR-APC, along with the ability to evolve oxygen, is further supported spectroscopically by absorbance and 77K fluorescence measurements. The 2D excitation/emission spectra (Fig. 2B) presents fluorescence at 750 nm (typical fluorescence emission of *C. thermalis* FR-PSII) when the sample is eluminated in the 550-600 nm range. The emission at 750 nm at these wavelengths further supports our claim that some connected complexes are present, since a large proportion of the absorbance in this area is coming from the phycocyanobilin pigments in the ApcB2 subunits of the FR-APC complex. We are aware that carotenoids present in PSII can contribute marginally to the absorbance of FR-APC at these wavelength, but the fluorescence intensity of the 750 nm cannot be explained only by the absorbance from FR-PSII, especially given the likely low amount of PSII present.

In terms of CryoEM data processing, we consider the box size to be appropriate to resolve the full complex. The extraction box size was 700 px (at a pixel size of 0.723 Å/px), which converts to 50 nm. The maximum expected size of the complex is 20 nm x 20 nm x 15 nm, therefore even with a box centered on the FR-APC cylinders, this should be large enough to resolve the full complex.

Nonetheless, due to the high interest of the field in understanding the phycobilisome connection points to FR-PSII, multiple attempts were made to resolve the density relative to the low occupancy FR-PSII, including local refinements masking the PSII dimers, 3D classification, and manual picking of particles that seem to present a connected complex with poor results.

We think that the lack of resolution for the PSII portion of the complex is due to multiple reasons, making the efforts in computationally improving the quality of the reconstruction futile: 1) the total particle count for the reconstruction presented in this paper is already low (~36k). 2) the proportion of particles with an intact complex is low, this can be seen by the absorption spectra of the sample as well as the fluorescence intensities of the 730 nm and 750 nm peaks at 77K. 3) We think that due to the loss of membrane integrity during the solubilization step and the consequent loss of spatial organization of the membranes, the connection of APC to PSII is weakened, leading to conformational heterogeneity and a general lack of rigidity of the complex.

Some 2D classes and *ab-initio* reconstructions present a diffused low intensity ESP density at the expected location of FR-PSII, which is well represented in the masks computed by cryosparc, but it is completely lost upon refinement due to the extremely low intensity of the signal.

The micrograph that was shown in the processing workflow was not representative of the average micrograph in our collection and has been replaced with a more suitable one. Unfortunately, we did not perform negative stain screening.

To acknowledge the limited evidence in favor of the presence of significant FR-PSII ESP density, a point the main text has been modified accordingly (page 5, lines 23-25):

“In addition, the map presents ESP density extending from the PB loop of ApcE2 that can be attributed to the presence of a subpopulation of particles with connected photosystems. Unfortunately, the limited number of particles available, the low percentage of connected complexes and heterogeneity prevent further local refinement of the photosystem portion of the map (fig. S4B). Attempts at resolving this density with local refinement and 3D classification yielded poor results (fig. S5).”

3. Page 6, “550 – 600 nm light, where the only species absorbing are the bilin pigments”. Please phrase this more carefully, as PSII does have absorbance in those wavelengths.

We thank the reviewer and we have rephrased this section for clarity. The new sentence is (page 7, lines 1-2): “When exciting the sample with 550 - 600 nm light, where the absorbance of the FR-APC dominates...”

4. Page 7. Please clarify how the torsion angle measurements were made. Which four atoms were used? Fig. 3A seems to indicate that four dihedral measurements were made, but the data show only two dihedrals. Furthermore, the measurements should all be provided as a SI table, including the measurements between ring B and C as a semi-control.

Figure 3A was clarified, now showing only one arrow per dihedral angle. The Methods sections relative to this measurement have been added (see Bilin angle measurements section in material and methods). We now provide all of the bilin angle measurements as a xlsx file in the SI.

5. Page 7-8, “...given that the loss of cysteine linkage in ApcD3 is only found in a few species ... suggests that ApcE2 is the main terminal emitter...”. Can this be supported by experimental data? If this is speculative, please clarify the wording accordingly.

We consider enough evidence to be available to suggest the role of ApcE2 as terminal emitter (as AcpE in WL-PSB complexes). In particular:

- The fluorescence emission of heterologously expressed ApcD3/ApcB2 heterodimers presents different fluorescence profile compared to the one of the intact complex (Soulier, 2021).
- The modeling of the FR-APC FR-PSII complex has suggested (both in this paper) and in previous ones that the chromophore in ApcE2 is closer to PSII compared with the one in ApcD3. Leading to more efficient excitation energy transfer from ApcE2.
- The bilin chromophore in ApcE2 is not only more planar than the one in ApcD3, but also presents full conservation of the missing Cys linkage in all FaRLiP capable species.

We thank the reviewer for pointing this out and we have improved this portion of the manuscript by more clearly stating the evidence for this claim. Moreover, we have made clear that contributions by other longer wavelength emitting subunits (ApcD3 and ApcF2) are possible and vary potentially between different far-red species.

The passages regarding this that were altered are the following (page 13, lines 1-3):

“Given that the loss of cysteine linkage in ApcD3 is only found in a few species (Ho et al., 2017) and that FR-PSII subunits are well conserved amongst each other, it is likely that the excitation injection point of FR-PSII from FR-APC is also conserved. This therefore suggests that ApcE2 is likely the main terminal emitter, also consistent with the bilin in ApcE2 being the most planar (Fig. 3B) and forming an ($\alpha\beta$) heterodimer with the most redshifted β -subunit ApcF2. This, however, does not exclude potential contributions by ApcD3 or ApcF2, and a degree of species specificity in the excitation energy transfer pathways to the photosystems in different far-red adapted strains.”

and in the discussion section (page 13, lines 30-33):

“Nonetheless, the difference in fluorescence emission profiles between ApcD3/ApcB2 and the intact FR-APC complex suggests that ApcE2 is the primary terminal emitter. The fluorescence of the ApcD3/ApcB2 monomer is much broader, whereas the emission from the intact complex is sharper and slightly blue-shifted (Fig. 2B). This indicates that while excitation energy may localize at both ApcD3 and ApcE2, the latter contributes more significantly to the final emission. Furthermore, the strong blue shift of ApcD3, with minimal spectral overlap with the fluorescence emission of the intact complex, suggests that during equilibration, excitation energy transfer is favored from ApcD3 to ApcE2 in comparison with the reverse direction.”

And (page 17, lines 3-5):

“in this model, the closest assigned Chl *f* pigment in FR-PSII to the terminal emitter bilin in ApcE2 is C507. Chl *f* C507 is at an edge-to-edge distance of ~ 31 Å from the ApcE2 bilin directly above CP43 and is ~ 36 Å away from the ApcE2 bilin between the two dimers, consistent with previous structural and functional studies (Gisriel et al., 2024; Mascoli et al., 2022). Moreover, the bilin pigment contained in ApcD3, previously proposed as a potential terminal emitter in

FR-APC, is located further away from any assigned Chl *f* in FR-PSII when compared with the bilin in ApcE2, suggesting the role of the latter as main terminal emitter in FR-APC.”

6. Page 8, Fig. 4. “Protein backbone in orange”. Remove ‘backbone’. Side chains are also shown.

Point addressed

7. Page 8, “when compared to the one in WL-ApcA”. How valid is this comparison? Please clarify this for the reader. My understanding is that the present FR-APC is from a different organism from both the previous FR-APC and the WL-APC being compared against. Please be careful, here and throughout the manuscript, to clarify if there are any sequence differences that should be considered alongside any structural comparisons. Show accompanying justification and data in the SI.

We have carefully reviewed the manuscript to make sure that comparisons are explained and justified. We consider this comparison of the far-red paralogue AcpD family subunits with AcpA to be the most appropriate, since ApcD2, ApcD3 and ApcD5 replace ApcA in FR-APC compared with WL-APC. All sequences present a high degree of conservation for far-red light organisms, and we consider, as previous literature on the topic does, the comparison between ApcD5 and ApcA subunits to be the appropriate one. We now have included a more in-depth comparison of the bilin pigments sites and altered the sentence to clarify for the reader (page 10, lines 16-18):

“To identify the amino acid residues responsible for red-shifting the absorption spectrum of α -subunits, far-red protein sequences were compared to their white light paralogues. ApcD5, ApcD3, and ApcD2 were analyzed in relation to ApcA rather than their paralogue ApcD1 because they occupy the same position structurally.”

8. Page 9, Fig 5. Panel E is mislabeled.

Fig. 4 and Fig. 5 of the main text have been reworked to include the improvements suggested by reviewer 3. The label to panel E has been fixed.

9. Page 9, the ApcB2-Cys75 and “loop between two helices” are not indicated in Fig. 5 despite the reference.

Point addressed, now the relevant portion of the ApcB2 subunit as well as the loop are shown and highlighted in the alignment panel.

10. Page 10 “... steric hindrance that would prevent ApcC from inserting...”. I echo my first critique that additional evidence is necessary to conclude that ApcC is natively excluded, not

just lost in the present sample. I am not fully convinced that the interactions shown in Fig. S2 are substantial enough to prevent the entire ApcC subunit from binding. By aligning the WL APC structure, the ApcC subunit does not appear to globally clash with FRL-ApcE2. On the contrary, many of the polar-polar interactions appear conserved. The clashes described in Fig. S2 could conceivably be accommodated by a local backbone shift. Furthermore, I reiterate my previous comment about the comparison between WL-APC and FRL-APC, that the organism and possible sequence differences needs to be clarified and considered. Please expand on this analysis, as it is a key point in the current manuscript.

We have, throughout the paper, softened the position on the absence of ApcC. We didn't consider this to be one of the crucial points of our manuscript, but just an observation in the structure that does not influence the main conclusions of our work. Therefore, we now propose both scenarios as possible throughout the text.

The following sentence has been modified (page 12, lines 3-12):

“As pointed out in the structural overview of the complex, the distal ApcC subunit, which in WL-APC is present between the third and fourth ring of the complex (Domínguez-Martín et al., 2022), is absent in this map of FR-APC (Fig. 6). In ApcE2 sequences, the loop interacting with the distal ApcC presents poor sequence conservation and some of the residues with bulkier side chains could create steric hindrance that would prevent ApcC from inserting in the distal site when compared with the white light paralogue ApcE (fig. S2). Moreover, even at high sigma values, no ESP density is present at the position normally occupied by ApcC, arguing against partial occupancy of this subunit.

Together these observations leave the possibility open that the absence of ApcC might be native rather than a purification artifact.”

And (page 5, lines 13-16):

“Due to its location, between 3rd and 4th ($\alpha\beta$)₃ trimers, the absence of the distal ApcC could be due to its loss during isolation, but the absence of any ESP density that could indicate partial occupancy, together with far-red light specific changes to the binding pocket, might imply that this subunit is natively absent from FR-APC cylinders.”

The other part of the question, relative to the comparison of FR and WL subunits is answered in point number 9. In short, the comparisons of subunits made throughout the text are relative to conserved features in far-red light adapted subunits compared with those in WL adapted subunits.

11. Page 10, regarding the Gln40 and Gln41 interactions. Firstly, the Gln40 and Gln41 conformations shown in Fig. 6 do not seem to match the conformation in the provided PDB. Please verify this point. Furthermore, the Gln conformations discussed are not the conformations shown in previous structures, both FR and WL. While it is true that the

orientation of the amide's carbonyl and amine groups are not easily differentiated structurally, the nearby ApcB2 Tyr87 seems to be a more likely proton acceptor for ApcC-Gln41 and a backbone carbonyl Asn110 seems to be a more likely proton acceptor for ApcC-Gln40. The bilin carbonyl seems a less likely proton acceptor for either of these Gln side chains. This is consistent with previous FR and WL structures, and the current FR-APC cryoEM data. Furthermore, the geometry to support the Gln-amine to bilin-carbonyl H-bonding is far from ideal for either Gln40 or Gln41, nor are they particularly supported by the cryo-EM data. Rather, consider that ApcC-Phe37 and ApcB-Tyr87 constrain the bilin ring.

We thank the reviewer for the in-depth analysis of the differences in the bilin sites in presence and absence of ApcC which has led us to reassess the interaction of ApcC with the ApcB2 bilin pigment. We have evaluated the proposed hydrogen bonds in both sharpened and unsharpened maps and even if the resolution is not high enough to exclude either of the two possible conformations, we consider evidence to be present to make the suggestions presented in the current manuscript.

Moreover, we have re-refined our structure based on the higher resolution map and realized that both bilin pigments in ApcB2 in presence and absence of ApcC are in the ZZZ_{asa} conformation, contrary to what we previously suggested.

Nonetheless, ring D of the bilin pigment in absence of ApcC is less planar than in its presence (Fig. 6). We consider ApcC Gln41 is likely to be interacting with the backbone carbonyl of Asn110, as suggested by the reviewer. On the other hand, the asymmetric density of the sidechain of ApcC Gln40 seems to suggest that the Gln nitrogen is facing the bilin keto group. We now include the suggestion made by the reviewer that ApcC Phe³⁷ also constrain the bilin site.

The sentence relative to the interaction of ApcC with the bilin pigment present in ApcB2 has been changed to (page 12- lines 16-18): "The presence of the hydrogen bonding residue ApcC Gln⁴⁰, together with the steric effect of ApcC Phe³⁷, ensures that the orientation of ring D is fixed and in a more planar conformation when compared to the ApcB2 bilin in the absence of ApcC (Fig. 6), likely affecting the absorbance spectrum."

12. Page 10 "bilin pigment is slightly stretched". Please provide a measurement and report the difference.

To follow on this point, we have made a systematic measurement of the bilin pigments stretching in our structure and realized that, even if minor stretches are present, the current resolution does not allow us to confidently measure this feature. This statement has been removed from the text.

13. Page 11, "interacts extensively with an ApcD5 subunit in the third ring...". This point is not

obvious in Fig. 1. In Fig S3, all the depicted interactions should have their side chains labeled. The H-bond distances should also be reported, either directly in the figure, or as a SI table. Some of the H-bonds, especially in panel C, appear too long. Overall, increased specificity of the data will strengthen the author's claim.

We thank the reviewer for pushing us to strengthen our analysis on this part. Fig. S3 has been modified accordingly, now the residues involved in the interactions are labeled and the hydrogen bond distances reported. Our statement on the interactions between the opposite ApcF2 subunits has been adjusted, since the high flexibility of the sidechains mentioned does not allow to assess their rotameric conformation. Nonetheless, we do still consider the distances to be compatible with this possibility, therefore we still mention this in the main text in the following form (page 13, lines 17-22):

“Moreover, the ApcF2 subunit may interact with the ApcF2 in the opposing cylinder. When the two cylinders are brought in proximity by the dimerization, the two ApcF2 subunits are in position for long polar sidechains to interact and provide stability to the complex. The high degree of flexibility of these sidechains, together with the poorer ESP density at the periphery of the map do not allow these interactions to be observed.”

14. Page 12 “which likely explains why its absorption profile is similar to its WL paralogue”. Please provide a reference for this.

References were added.

15. Page 12, “This evolutionary adaptation indicates ...”. I am under the impression that the phycobilisome has several morphologies across different cyanobacterial species and that this is not limited to WL vs FRL/LowLight. Is it technically correct to say that the difference in morphology is directly caused by the evolutionary pressures of different photo-niches? Please expand on this point for the reader, providing references. I also note that this point is referenced in the Abstract and would likewise benefit from additional discussion of the literature.

Phycobilisomes exhibit diverse morphologies and evolutionary variability in both absorbance profile and size. However, to date, the smallest known phycobiliprotein antenna system is found in far-red light-acclimated organisms. FaRLiP-capable organisms share a highly conserved gene structure, resulting in the smallest phycobilisomes among all cyanobacteria.

The reduction in antenna size is not the only adaptation shaped by the evolutionary pressure of a visible light-limited photo-niche. Instead, (at least) four interrelated factors contribute to this adaptation:

- 1) The red-shifting of bilin pigments.

- 2) The reduction in overall antenna size, allowing for tighter packing of thylakoid membranes and, consequently, a higher density of photosystems.
- 3) These photosystems contain a distinct set of longer-wavelength chlorophylls, extending absorption beyond that of the red-shifted bilins.
- 4) The finding by Mascoli et al. that FR-APC accelerates excitation energy trapping by FR-PSII when connected.

The statement in the main text has been expanded and referenced properly to address all of these contributions (page 15, lines 7-30):

“The structure of the bicylindrical FR-APC prompts questions on the importance of antenna size in far-red light photoacclimation. Phycobilisomes present a range of morphologies and evolutionary variability both in terms of absorbance profile and size, but to date, the smallest phycobiliprotein antenna system to be characterized is that found in far-red light acclimated organisms (Bryant & Gisriel, 2024). The FR-APC bicylindrical complex provides an extensive enhancement of the absorption cross-section in the far-red for FR-PSII, providing 24 far-red absorbing bilin pigments compared to the 10 long-wavelength chlorophyll pigments found in a FR-PSII dimer (Nürnberg et al., 2018). Nonetheless, as previously pointed out, the overall number of bilin pigments in FR-APC is much lower than in phycobilisome structures found in cyanobacteria grown in white light, where hundreds of bilins are present (Bryant & Gisriel, 2024).

Why would a photosynthetic organism shaded from visible light sacrifice most of its light harvesting capacity? The presence of smaller phycobilisomes allows cells to increase the thylakoid and thus photosystem density, an observation supported by the reduced distance between thylakoid membranes during far-red light photoacclimation (MacGregor-Chatwin et al., 2023; Li et al., 2016). This evolutionary adaptation might indicate that space optimization within and between the membranes is preferred over antenna size in far-red light adapted thylakoids. This reorganization could be advantageous by increasing the Chl *f*-to-bilin ratio, thereby enhancing absorbance at wavelengths beyond 710 nm.

Moreover, the differences in absorption spectra between far-red α - and β -subunits imply that the spectral range covered by the FR-APC evolved to be broader than the WL-PBS complex, even though WL-PBS are bigger and structurally more complex. The steeper absorption gradient of the bilins in FR-APC relative to WL-APC may result in a more efficient excitation transfer from FR-APC to FR-PSII, consistent with the enhanced trapping rate of FR-PSII when the FR-APC is connected (Mascoli et al., 2022).”

16. Page 14, Fig. 7. Please emphasize for the reader that these models are based on the previous in situ results. Please also provide their PDB code in the figure description.

Point addressed

17. Page 14. “not parallel but at an angle of roughly $\sim 20^\circ$ ”. Zhang et al. 2024 report a 6° difference. Please clarify this discrepancy. (Zhang et al. 2024 report a 25.3° twist of the APC cores, a measurement that should similarly be reported on the current FR-APC structure).

In the Zhang et al. 2024 paper, no method on how the 6° angle provided is presented. For this reason, we recalculated the angle based on the C2 symmetry planes of FR-APC with respect to the one of the FR-PSII dimers, leading to an angle of $\sim 19^\circ$. Since we fitted our model on their structure the expected angle is the same.

In the same way no method is presented for the calculation of the twist angle between the two FR-APC cylinders in Zhang et al. 2024. To do that we calculated the geometric centers of each of the four ab3 trimers and interpolated two axes. The angle between the two axes is at about $\sim 30^\circ$ and is now reported in the main text. The methods section has been updated accordingly.

For clarity the sentence has been modified to (page 16, lines 13-16): “the C2 symmetry planes of FR-PSII and bicylindrical FR-APC are not parallel but at an angle of roughly $\sim 20^\circ$, suggesting that the two cylinders of the FR-APC are likely to have two independent docking sites on FR-PSII dimers”

Moreover, the angle of twist of the FR-APC cylinders is reported as follows (page 5 lines 3-7): “The map presents density to fit two antiparallel far-red light adapted allophycocyanin cores at an angle of $\sim 29^\circ$ (see materials and methods)”

18. Page 17. “GC-Fourier Shell Correlation”, Typo. GS is correct, but “gold standard” is better.

Point addressed

19. Page 23. “script ... is available upon request”. Basic details about what the script does and how it is used should be described in the methods section. See critique #4.

The details on how the calculation is performed are reported in the method section.

20. Table S1. The final particle count does not match what is in the Methods section.

The particle count in the method section was wrong due to a typo and has been rectified.

21. Table S2. Please review existing cryo-EM literature to see how these values should be reported and which values are needed. Clash score, for example, is missing from your table.

Table S2 has been rectified according to what is standard in the literature. The clash score has been added.

22. Fig. S1. The 2D classes are mislabeled.

Point addressed.

23. Fig. S1. Please review recent cryo-EM literature and include standard data. Several data are missing, for example, the FSC curve (the cFSC is not a replacement, as it evaluates orientation bias), map-model FSC curve, angular distribution plot, a clearer local resolution map, etc. Furthermore, the workflow diagram should be much more substantial.

Point addressed. Recent CryoEM literature has been reviewed and all of the information has been added to Figure S1. The information missing is still present in the PDB validation report. Moreover, the workflow diagram has been extended, in the first version it was simplified for clarity.

24. Minor grammatical errors are found throughout.

We reviewed the paper and removed the errors we were able to find.

Minor Suggestions:

1. Page 8, Fig 4. Labeling ApcE as WL and ApcE2 as FR might improve readability.

We understand the point that the reviewer is trying to make, but we believe that adding WL and FR to the protein labels may introduce a false understanding that there is an equivalent FR gene is one is labeled WL (or vice versa). We have attempted to make clear throughout the text the distinction between which proteins are only expressed and present in the FR APC or only in WL and used the gene notations conventional in the literature.

2. Page 8, Fig 4. Throughout the manuscript and SI, coloring/formatting the sequence alignment using ClustalX formatting is more informative and more consistent with existing literature (especially outside the photosynthesis field).

We provide the full alignment file as a supplementary material, so that each reader can delve into the protein sequences. We appreciate the reviewer's prompt that other outside of the photosynthesis field would be interested in the manuscript. However, for clarity within the

figures we have decided to leave the other figures presented this way to improve readability. We are aware that alignments, when colored, are often reported using the ClustalX formatting to highlight amino acid differences, but considering the focus of the analysis on specific conserved far-red light specific changes, we consider the presentation with boxes highlighting them more appropriate for the scope of this paper.

3. SI, Fig. S3. In panel B, the intra-helix H-bonds are unnecessary and confuse the inter-subunit interactions being discussed.

Fig. S3 has been reworked to improve readability as per major comment 13, we also added the labels of the interacting amino acids and H-bond lengths.

4. Page 9, Fig 5. In panel A, ApcA-Asn71 appears to H-bond with the bilin from this angle, even though this is not the case. Asn71 can be hidden to avoid confusion. Furthermore, consider showing H-bonding using dashed lines.

Fig. 4 and Fig. 5, as well as the analysis of the conserved changes of the bilin sites have been reworked in accordance with the comments by reviewer #3. We hope that the new version is less confusing and more informative than the previous one.

5. Page 9, Fig 5. Panel A is improperly cropped.

Fig. 5 has been also reworked.

Reviewer #2 (Remarks to the Author):

Summary

The MS from Consoli et al describes the cryoem structure of a bicylindrical APC induced by far red (FR) light from a C. Thermalis. This is a nice addition to the field which complements and improves the description of light harvesting systems in under FR conditions, a subject of considerable general interest. Compared to previous studies a more intact form of this APC was obtained, which contains the “missing” ab trimer and retains the overall bicylindrical organization. The photophysical properties of this sample are probably a better representation of the in vivo APC than previous structures. The presentation of the work is good, and the quality of experimental data is high. In general, I think the MS should be accepted with minor modifications only.

We thank Reviewer 2 for the kind comments. We want to inform the reviewer that we have now obtained a slightly better resolved map (2.51 Å) of the complex and that we have redeposited it under the same accession code. Here is a link to the new maps and model:

https://drive.google.com/drive/folders/1GL9zWQVeEZr_DifUfGSyLxPhKA8OQLGR?usp=drive_link

Minor points:

1. Line 73 contains “Click or tap here to enter text.”.

Point addressed, the reference managing software malfunctioned.

2. In the provided model Chain L/I Ala67 seems to be a different amino acid.

We also noticed a prominent density at this position, but after accurately checking the multiple sequence alignment we consider Chain L/I at position 67 to be an alanine. The abnormal density at position 67 is not present in all of the other copies of the ApcD5 subunit. We thank the reviewer for the concern.

3. Fig. S1 – the use of C2 symmetry is mentioned in the text, but the symmetry used in refinement should be indicated at the appropriate steps in section A.

The processing workflow has been improved and the use of C2 symmetry is now included in Fig. S1.

4. Overall clashscore and molprobity score should for the final model should be included in table S2.

The scores are now included in table S2.

Reviewer #3 (Remarks to the Author):

Summary

The manuscript by Consoli et al. reports a cryoEM structure at 2.6Å resolution of far-red allophycocyanin (FR-APC) in a bicyclic antenna complex. The contribution of this work to the field/topics of light-harvesting and photosynthesis in cyanobacteria can be summarized in two aspects. First, this is the first intact cryoEM structure reported for the FR-APC complex where all four expected phycobiliprotein trimers are present. The remarkable conformational diversity of PCB revealed by the FR-APC structure allows in-depth examination of the molecular basis underlying the PCB-based far-red absorption. Second, the authors presented a model of FR-APC docked onto FR-PSII using the published cryoET data as a guide. In the absence of an experimental structure, such a model provides an important structural glimpse into the directed energy transfer from the bilin pigments of APC to the chlorophyll f in PSII. Specifically, this docking model suggests that ApcE2 (not ApcD3) is the main terminal emitter in FR-APC. Overall, this manuscript is well written, and the findings are interesting. However, some weaknesses need to be addressed before publication. Since the bilin conformational diversity is the forte of this paper, it requires in-depth analysis of the protein-chromophore interactions beyond the planarity analysis (Fig. 3). Specifically, plain description of the protein environment is not sufficient for revealing the structural basis of far-red absorption of PCB. Given the excellent resolution and diverse configurations of PCB, this cryoEM structure offers a unique opportunity to identify key interactions (again beyond planarity) responsible for extending the absorption spectra of PCB to the red and far-red regions. See below for comments.

We thank the reviewer for the insightful comments on the conformation of the bilin chromophores in our structure, these comments have been pivotal in improving our manuscript and addressing some of the problems in the modeling of the chromophores. In the new version of the paper, the bilin densities as well as their chemical environment has been thoroughly analyzed, leading to two major findings.

1) A portion of the conformational diversity in the bilin pigments was due to artifacts in the refinement of the structure. The conformation of the bilin pigments is now discussed and analyzed in the text.

2) We realized that the ApcF subunit (in purple throughout the manuscript), corresponds to the structurally novel subunit ApcF2, and was badly annotated. The gene coding this subunit is positioned outside of the FaRLiP cluster and we mistakenly considered it to be a WL subunit. We have reworked the manuscript to include a description of the ZZZ_{ssa} conformation of the bilin pigment, together with a discussion of the differences between ApcF2 and ApcE2.

We have improved the overall presentation of the manuscript and hope the reviewer still considers the findings of the paper interesting, even though the diversity of the bilin pigments is less than previously presented. Finally, we want to inform the reviewer that we have now obtained a slightly better resolved map (2.51 Å) of the complex and that we have redeposited it under the same accession code. Here is a link to the new maps and model:

https://drive.google.com/drive/folders/1GL9zWQVeEZr_DifUfGSyLxPhKA8OQLGR?usp=drive_link

Major points:

1. Please provide the focused (masked) density maps (in supplemental materials) to justify the structural interpretation (Z/E and syn/anti) of all PCBs in the FR-APC complex.

The focused ESP CryoEM map for each of the bilin pigments present in the FR-APC structure is now provided in the supplementary materials (fig. S5).

2. The FR-APC complex features PCB in diverse configurations (Fig. 4 and Fig. 5). Judging from the views provided, ApcE features PCB in ZZZ,ssa while PCB in ApcE2 adopts an all-Z,syn configuration (Fig. 4). Both ApcA and ApcD5 show PCB in ZZZ,ass while ApcD2 and ApcD3 show PCB in ZZZ,asa, which is typically found in phycobiliproteins (Fig. 5). Such diversity is highly interesting and needs better illustration. A multi-panel figure that shows the PCB structures (in a face-on view NOT side view) from different subunits of FR-APC (perhaps including WL-APC) would be helpful to highlight such diversity and facilitate direct comparisons.

Prompted by the reviewer question, after a careful analysis of the far-red conserved changes and of the density ESP map (both sharpened and unsharpened) for all of the bilin pigments in this structure, we came to the conclusion that some of the PCB conformations presented in the previous draft of the paper were unsubstantiated by the presence of amino acids conserved changes to justify them.

In particular, we previously assigned ApcE2 to be in the all-Z,syn configuration based on the phenix real space refined structure, but thanks to the reviewer's comment and a careful analysis of the environment in comparison with previous ApcE structures, we now consider The PCB pigment in ApcE2 to be in ZZZ-ssa conformation like in ApcE.

Concerning ApcD5 we also have reviewed our assignment. Due to the absence of conserved changes, we have re-refined the pigments in the ZZZ,asa configuration. This error was partly due to the observation of previous structure in the literature, since some of them feature the ApcA PCB pigment in the ZZZ,ass configuration.

Finally, we realized the presence in our structure of the far-red specific ApcF2 novel subunit (in place of ApcF). This subunit features a PBC chromophore in the ZZZ,ssa configuration (as in ApcE) and no Cys linkage of the chromophore.

The figures presenting the bilin sites (Fig.4 and Fig.5) have been completely reworked and now feature aligned face-on views of the compared pigments, including the WL version of the environment.

In Fig. 4 we compare the differences in the sites between ApcE and ApcE2 and ApcF and ApcF2, as well as the differences that might further redshift ApcE2 compared with ApcF2 to suit its role as terminal emitter.

3. It is well known that other than the bilin planarity (depicted by the twist angles between pyrrole rings in Fig. 3), specific protein-PCB interactions play critical roles in spectral tuning. Please analyze and discuss what unique protein-chromophore interactions in the chromophore-binding pocket, including ionic interactions, geometric factors or presence of specific Cys anchors, contribute to the tuning of the absorption spectra, in particular, the far-red absorption. Plain description simply does not do the job. Further analysis with tabulation and plots (against the peak wavelengths) are needed. Discussion may also cover the current understanding of spectrum-tuning mechanisms obtained from other bilin-binding proteins, e.g. cyanobacteriochromes (CBCRs).

We have improved the section that discusses the conserved changes and their potential effect on the site energy of the bilin pigments (page 9-12). We have cited the appropriate literature that presents the absorption and low temperature emission of the various AB heterodimers featured in this complex and tried to rationalize the conserved changes with the shifts in color. Moreover, we also made a comparison with the low light expressed AB heterodimer, AcpD4/ApcB3. We report that the sites are mostly conserved, suggesting once again that the terminal emitter of the FR-APC complex is likely to be ApcE2.

4. The modeling of the APC-PSII complex is interesting and significant. Perhaps, these texts are better suited for the Results section. An alternative would be to combine Results and Discussion as one section.

We consider the placement of the modeling of the APC-PSII model to be appropriate in the discussion, since we only fitted a structure already available in the literature. Moreover, this might improve readability of the manuscript.

In case the reviewer is keen on making this change we can accommodate it and rework the manuscript to incorporate results and discussion if the editor agrees.

Minor points:

1. Please mark the C2 symmetry in Fig. 1B (right panel).

The C2 symmetry is now represented in Fig. 1B

2. Line 248-251: the statement that “The bilin pigment in ApcE2 is non-covalently bound and almost completely planar, pushing the physical properties of phycocyanobilin towards the maximum red-shift attainable by this pigment (Staheli et al., 2021; Miao et al., 2016).” is not accurate. The maximum red-shift reported for PCB has exceeded 740nm for a far-red cyanobacteriochrome exhibiting an all,Z-syn conformation covalently anchored to a Cys residue (Rockwell et al, 2016).

We have removed this statement even though we have realized that the PCB in ApcE2 is in a ZZZ,ssa conformation. Nonetheless, we consider interesting the possibility that PBP might not be capable of harboring all Z-syn chromophores or that PBC in this conformation are not

suitable for efficient energy transfer in antenna systems. That would make AcpE2 the longest available emitter in APC complexes.

Modified Figures:

Fig. 1 – Structure of the bicylindrical FR-APC and positions of the bilin pigments.

A) View of the dimeric FR-APC map from a tilted perspective facing the 1st ($\alpha\beta$)₃ from the cytoplasmic side. The map presents clear ESP for all the known FR-APC subunits, color key at the bottom left of the panel. B) Location of the phycocyanobilin pigments in the bicylindrical FR-APC complex as viewed from the membrane plane and from the cytosolic side of the membrane. The pigments are also colored according to subunit coordination in panel A. C) Side view of a single FR-APC cylinder with ApcE2 colored in orange and every other subunit transparent. D) Separate representation of the four ($\alpha\beta$)₃ trimers, also colored according to the color key in panel A. It is important to note that the ApcE2 subunit is only represented in the second trimer.

Fig. 2 - Analysis of the planarity of the bilin pigments in FR-APC and WL-APC

A) Representation of an unbound planar phycocyanobilin pigment with the measured torsion angles indicated. B) Variation of bilin planarity in FR-APC. The x-axis reports the degrees off plane of ring A with respect to ring B, while the y-axis reports the degrees off plane of ring D with respect to ring C. C) Variation of the planarity of the bilin pigments in WL-APC from *Synechocystis sp.* PCC 6803 (7SC7). The x-axis reports the degrees off plane of ring A with respect to ring B, while the y-axis reports the degrees off plane of ring D with respect to ring C.

Fig. 3 – Protein environment of the bilin in the ApcE2 and ApcF2 subunits.

A) Chemical environment of the bilin pigment in the ApcE protein (7SC7). B) Chemical environment of the bilin pigment in the ApcE2 protein. C) Sequence alignment between selected ApcE and ApcE2 sequences. The light blue box highlights the discussed conserved loss of Cys between WL and FR paralogues. D) Chemical environment of the bilin pigment in the ApcF protein (7SC7). E) Chemical environment of the bilin pigment in the ApcF2 protein. F) Sequence alignment between selected ApcF and ApcF2 sequences. The phycocyanobilin pigments are represented in blue, ApcE in brown, ApcE2 in orange, ApcF in purple, ApcF2 in violet, and hydrogen bonds in yellow. The light blue boxes in the sequence alignments highlight the conserved loss of Cys in the FR paralogues.

Fig. 4 – Changes in the bilin sites of ApcD2, ApcD3 and ApcD5 compared with the WL paralogues

Comparison of the chemical environment of the bilin pigment in the ApcA protein (A) against the ApcD5 (B), ApcD2 (C), and ApcD3 (D) proteins. The bilin pigments are always represented in blue. E) Sequence alignment between ApcA and ApcD5, the light blue box highlights conserved changes in the sequence between white light and far-red paralogues and the pink box highlights the loop deletion discussed in the text.

Fig. 5 – Comparison between the chemical environments of different ApcB2 subunits in the FR-APC complex.

In the center, one cylinder with ApcE2 is represented in orange, ApcC is represented in yellow and two ApcB2 subunits in green. A) Proximal ApcB2 bilin binding site in the presence of ApcC. ApcB2 is represented in green, ApcC is represented in yellow, and phycocyanobilin is represented in blue. B) Distal ApcB2 bilin binding site in the absence of ApcC. ApcB2 is represented in green while phycocyanobilin is represented in blue. The position of the proximal bilin site compared is represented in transparent pink.

Fig. S1 - CryoSPARC workflow for the SPA structure determination of the FR-APC complex

CryoSPARC workflow for the determination of the map of the FR-APC complex.

Fig. S3 – Interactions at the dimer interface between the two FR-APC cylinders.

A) The interaction of ApcE2 (in orange), with an ApcD5 (in teal) subunit of the third ring of the opposing cylinder are analyzed on the panel on the right. B) ApcD2 (in pink) interactions with an ApcB2 (green) of the third ring and an ApcD5 (in teal) subunit of the fourth ring of the opposing cylinder are analyzed in the panel on the right. On the left of panel A and B the network of the interaction between monomers of the bicylindrical FR-APC is represented. Subunits are represented as circles of dimension proportional to their length in amino acids. Interaction of α and β subunits to form ($\alpha\beta$) monomers are represented as dashed lines, while inter-cylinder interaction are represented by black solid lines.

Fig. S4 - CryoSPARC workflow for the SPA structure determination of the FR-APC complex

A) Local resolution map of the FR-APC complex colored according to the colorbar on the side. B) GS-FSC plot of the resolution of the complex. C) cFSC plot of the resolution of the complex. D) Fourier sampling of the final reconstruction. E) Plot of the relative signal vs the viewing direction obtained with Cryosparc's orientation diagnostics tools. F) Guinier's plot of the FR-APC complex. G) Viewing direction distribution. H) Local resolution map at FSC = 0.5

Fig. S5 – ESP density of each of the CYC pigments present in the FR-APC complex

ESP density of the bilin pigments discussed in the paper. The ESP density and the bilin models are represented in the same colors used throughout the main text, each column of pigments represents the bilins contained in an alpha beta trimer, as represented in Fig. 1, the 3 rows, composed of 2 bilins each, represent each a set of 4 monomers aligned with respect to the position of the complex and of the membrane.

Fig. S6 – Local refinement mask and results.

A) Original map. B) Mask used to subtract from the particle set that covers the FR-APC cylinders. C) Example of a local refined map at the putative Photosystem II region showing only featureless blobs. To try and resolve the low-quality density that was observed on the membrane-side of FR-APC, multiple rounds of local refinements with different masks, with and without symmetry, were done. Briefly, masks of the APC cylinders were created using segger in ChimeraX. They were padded, dilated, and used to subtract the APC cylinders density from the particle set (symmetry expanded where appropriate) in a particle subtraction job in CryoSPARC (Fig. S5B). The resultant particle set was used to reconstruct an initial density, which was then fed into local refinement jobs. Unfortunately, none of the attempts were fruitful in resolving the putative Photosystem II density. Only featureless blobs were observed.

Reviewer #1 (Remarks to the Author):

The manuscript is much improved and all my comments have been adequately addressed. The additional analysis and discussion prompted by reviewer 3 is an excellent addition. I have a few minor points that should be addressed, but overall, the manuscript is in good shape.

We thank the reviewer for the kind comments, we have gone through the main text and hopefully addressed all of the new points presented.

1. Introduction final paragraph: “density localized where PSII is expected to bind.” Some of the 2D classes shown in Fig. S1. are data in support of this point. This could be mentioned, and an enlarged figure of a 2D class can be shown in Fig. S6.

As suggested, an enlarged version of the 2D classes is now present as panel D in Fig. S6, reference to the two figures has been added to the sentence.

2. Results paragraph 1: The $\sim 30^\circ$ angle measurement is a good addition but please modify the wording to make it clearer for a new reader.

We have modified the sentence slightly to make the wording clearer for a new reader. A more in depth explanation is provided in the Methods section. The sentence is now “...allophycocyanin cores at an angle of $\sim 30^\circ$ to each other”

3. I like the changes made to Fig. 4 and 5 based on reviewer 3’s feedback, however the side view is still of interest, especially since portions of the discussion emphasize the planarity. I suggest the original figures be included as SI, perhaps in a single composite figure omitting the sequence alignment but showing both perspectives of the bilins.

The old figure presented some model errors that have been rectified and has therefore not been included.

4. Fig. 6. ApcC Gln40 may be a typo as ApcC Gln41 is closer to Tyr91. Furthermore, in the updated pdb file the ApcC Gln41 amide orientation appears flipped relative to what is shown in Fig. 6. The software can sometimes do this without a clear warning, so please confirm the orientation is as intended by the authors.

We thank the reviewers for having caught this mistake Gln40 is not reported correctly both in the main text and in the figure label.

5. Right before Fig. 7, the first in situ seems to be redundant.

The first in situ has been removed

6. If the authors see fit, there may be some opportunity to improve the wording of the final few paragraphs of the Discussion, which are somewhat repetitive.

We agree with the reviewer and the wording has been slightly changed to improve the flow and remove repeated words

“From an experimental point of view, this behaviour could explain the inability to resolve structures of APC-PSII complexes via CryoEM single particle analysis, as membrane solubilization treatments would likely destroy the supercomplexes and consequently cause the disconnection of the allophycocyanin cores from the photosystems. This emphasizes the need for alternative characterization strategies, with cryo-electron tomography being an obvious approach³³.

This work highlights the importance of understanding Chl *f*-based photosynthesis at a scale that goes beyond individual complexes and their interactions, extending to the organization of the thylakoid membranes and *in situ* photochemistry. Investigating excitation energy transfer networks in the mesoscale will lead to a better understanding of the relationship between antenna size, photosynthetic efficiency, and photodamage when comparing different evolutionary adaptations to far-red light and other ecological niches.

Ultimately, this study illustrates the evolutionary trade-offs that allow efficient management of excitation energy transfer by far-red light adapted antenna. The insights obtained should help with the assessment of the feasibility, requirements, and limitations in the development of applications of far-red photosynthesis.”

Reviewer #2 (Remarks to the Author):

The authors addressed all of my concerns and I support the publication of the manuscript.

Reviewer #3 (Remarks to the Author):

The revised manuscript has properly addressed my questions/concerns from the first round of review. However, I do see extensive responses to the reviewer #1's questions, which appear critical but fair. To this end, I would support the publication of the revised manuscript if reviewer #1 is satisfied with the responses from the authors.